# DYNAMIC ELECTROENCEPHALOGRAPHY REPRESENTATION LEARNING FOR IMPROVED EPILEPTIC SEIZURE DETECTION

## ABSTRACT

Epileptic seizure is an abnormal brain activity that affects millions of people worldwide. Effectively detecting seizures from electroencephalography (EEG) signals with automated algorithms is essential for seizure diagnosis and treatment. Although much research has been proposed to learn EEG representations for seizure presence detection, most of these approaches are not suitable for seizure onset detection, which involves identifying the specific timestamps of seizure onsets. Several studies have been conducted to address this issue and investigate the onset detection or real-time detection, providing fine-grained insights into EEG signals. However, these studies often overlook the temporal correlation across EEG samples or fail to model the dynamics of brain activities. In this work, we introduce the Dynamic Seizure Network, a unified framework for EEG representation learning on both detection task and real-time detection task, which efficiently captures the *dynamic* dependencies for *real-time* seizure detection. Theoretical analysis and experimental results on three real-world seizure datasets demonstrate that our method outperforms baselines with low time and space complexity. Our method can also provide visualizations to assist clinicians in localizing abnormal brain activities for further diagnosis.

## 1 INTRODUCTION

Epilepsy is the most common neurological disease affecting around 50 million people worldwide (WHO, 2023). Epileptic seizure, an important characteristic of epilepsy, is caused by the abnormal surge of electrical activity in the brain and leads to a range of symptoms such as uncontrollable muscle movements and loss of consciousness.

Electroencephalography (EEG) is a standard tool for epileptic seizure monitoring, diagnosis, and analysis. It uses non-invasive electrodes placed on the scalp to measure and record the electrical activities of the brain. Figure 1 (a) illustrates the placement of electrodes. Then, professional neurologists will examine the recorded EEG signals to detect, classify, or locate abnormalities related to epileptic seizures. However, this manual analysis is labor-intensive and time-consuming, making it unviable for real-time seizure monitoring and detection. Therefore, it is essential to develop automated algorithms for efficient epileptic seizure detection.

In recent years, deep learning-based methods have been proposed for automated seizure detection (Figure 1 (b)). Dense-CNN (Saab et al., 2020) and CNN-LSTM (Ahmedt-Aristizabal et al., 2020) utilize convolutional networks to extract cross-electrode features from the EEG records. Tang et al. (2022) constructs two graph structures to capture brain connectivity. TSD (Ma et al., 2023) proposes using the self-attention mechanism to detect epileptic seizures. Methods designed for multivariate time-series modeling can also be adapted to model EEG signals. For example, STGCN (Yu et al., 2018) and MTGNN (Wu et al., 2020) model the temporal and cross-channel information with convolutional networks. CrossFormer (Zhang & Yan, 2023) and SageFormer (Zhang et al., 2023) respectively extend self-attention mechanism with cross-channel attention and graph neural networks.

While insightful, these methods can only detect the presence of seizures but fail to identify the specific timestamps when seizures start. To address this limitation, some studies (Tang et al., 2020; Thyagachandran et al., 2020; Wang et al., 2021; Shama et al., 2023) propose to investigate an improved seizure detection task, namely the *onset seizure detection* task, where the model predicts a

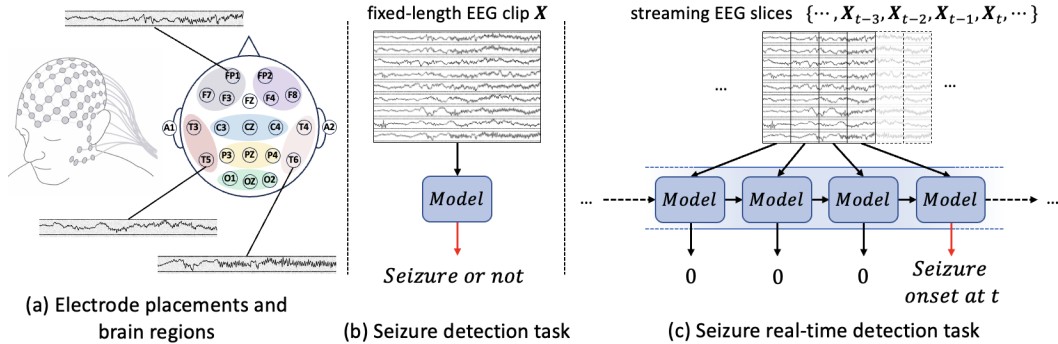

fixed-length EEG clip **X**

streaming EEG slices $\{\cdots, \boldsymbol{X}_{t-3}, \boldsymbol{X}_{t-2}, \boldsymbol{X}_{t-1}, \boldsymbol{X}_t, \cdots\}$

*Model*

*Seizure or not*

0    0    0    *Seizure onset at t*

(a) Electrode placements and brain regions

(b) Seizure detection task

(c) Seizure real-time detection task

Figure 1: (a) An illustration of EEG electrode placement in the standard 10-20 system. Each color denotes a brain functional region. (b) The seizure detection task. (c) The real-time detection task where inputs are streaming and the model identifies the specific timestamps of seizure onsets.

seizure/non-seizure label for each timestamp. They also evaluate the latency between the detection and ground truth onsets, providing more fine-grained insights into EEG signals.

Nevertheless, these existing studies on the onset detection task also have certain limitations in the context of *real-time seizure detection* (Lee et al. (2022), Figure 1 (c)), where the EEG samples are input consecutively in chronological order. For example, Tang et al. (2020); Thyagachandran et al. (2020); Wang et al. (2021) treat consecutive inputs as individual samples and neglect the temporal correlations across the samples. Shama et al. (2023) overlooks the causal restriction by accessing information from the future timestamps, resulting in information leakage. These shortcomings may restrict their applicability in real-time seizure monitoring. Lee et al. (2022) further proposed to capture the correlation among streaming EEG samples. This method has better availability in real-world clinical contexts, owing to the ability to detect seizures from real-time inputs with lower latency and resource utilization. However, this study primarily focuses on modeling the local dependencies within each sample but fails to further investigate the dynamic nature of brain activities.

Moreover, these methods either focus on modeling long-term dependency with self-attention mechanisms (Ma et al., 2023; Zhang & Yan, 2023; Zhang et al., 2023) or emphasize capturing the dynamics state of the objects with recurrent networks (Tang et al., 2022; Chen et al., 2022; Lee et al., 2022). Consequently, they tend to overlook the dynamic brain connectivities or have limited capabilities in localizing abnormal brain activities. None of the aforementioned works have successfully integrated both advantages together into a unified network while maintaining effectiveness and efficiency.

To this end, we aim to design a unified framework for both the seizure detection and the real-time detection tasks, which efficiently captures the dynamic dependencies within streaming EEG signals for real-time seizure detection. Specifically, we propose a recurrent attention module to model the evolutionary brain states from streaming EEG signals and a correlation learning module to capture the dynamic connectivities among brain regions, ultimately introducing a framework named **D**ynamic **S**eizure **N**etwork (DSN). Experiments on three real-world datasets on both the conventional seizure detection task and the onset detection task demonstrate the efficacy and efficiency of the proposed method. In summary, our main contributions include:

- We design a unified framework DSN for both the seizure detection and the real-time detection tasks, which is capable of capturing the dynamic dependencies from real-time streaming EEG signals and generating fine-grained predictions. We propose two metrics, *diagnosis rate* and *wrong rate*, to better quantify the detection latency under streaming context.

- We propose an evolutionary state modeling module to integrate the attention and recurrent networks for simultaneously localizing brain abnormalities and modeling evolutionary activities, which efficiently handles the streaming EEG signals with linear time complexity. We also employ a dynamic brain region correlation modeling module to adaptively learn both the dynamic and static interconnections among brain regions.

- Experiments on both detection task and onset detection task on three real-world clinical datasets substantiate the superiority of the proposed method. Theoretical analysis and technical experiments demonstrate the effectiveness and efficiency of DSN on the onset detection task. Visualization studies further illustrate the interpretability of our approach.

## 2  METHODOLOGY

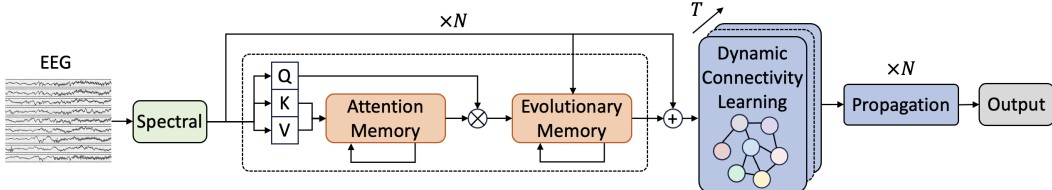

Figure 2: Overall architecture of DSN.

### 2.1  PROBLEM FORMULATION

EEG signals are recorded by the electrodes positioned on the scalp of the patients[1]. Each electrode records the voltage readings from a specific region of the scalp, implicitly reflecting the activities of the corresponding brain regions. Each record contains multiple time series recorded by different electrodes and can be formulated as a multivariate time series. Our research encompasses two tasks: the seizure detection task and the real-time detection task.

**Seizure Detection Task** is a classification task to identify the occurrence of an epileptic event within an input EEG clip $\mathbf{X} \in \mathbb{R}^{W \times C}$, where $C$ is the number of channels (i.e. electrodes) and $W$ is the length of the window (e.g. 30s). The label of the detection task is a binary scalar indicating whether any seizure event happens within this clip, i.e. $y^{det} = 1$ stands for seizure and $y^{det} = 0$ otherwise.

**Real-time Detection Task** is defined in a streaming manner. The inputs are EEG slices $\{\mathbf{X}_t \in \mathbb{R}^{S \times C} | 0 \leq t\}$ obtained by slicing the raw EEG records into non-overlapping $S$-seconds windows, where $S$ is much shorter than $W$ (e.g. 1s). These slices are sequentially input into the model in chronological order. The labels are binary values $\{y_t^{onset} \in \{0, 1\} | 0 \leq t\}$ indicating the presence or absence of seizures within each timestamp $t$. Compared with the conventional seizure detection task, the real-time detection task identifies not only the presence of seizures but also the precise moments of seizure events. Moreover, the sequential input of EEG slices in chronological order enables the modeling of long-term evolutionary patterns.

Furthermore, to quantify the models' timeliness in detecting seizure events, we introduce two new metrics, namely, the *Diagnosis Rate* and the *Wrong Rate*. Specifically, we define truth onset events as $\mathcal{E}^T = \{t | y_{t-1}^{onset} = 0, y_t^{onset} = 1, 1 \leq t\}$ and predicted onset events as $\mathcal{E}^P = \{s | \hat{y}_{t-1}^{onset} = 0, \hat{y}_t^{onset} = 1, 1 \leq t\}$. Then we define the diagnosis rate $Dr(k)$ and the wrong rate $Wr(k)$ as:

- **Diagnosis Rate-k** stands for the proportion of correct predictions within $k$ seconds since truth events happen. Formally,

$$Dr(k) = \frac{1}{|\mathcal{E}^T|} \sum_{t \in \mathcal{E}^T} max_{0 \leq s < k} \hat{y}_{t+s}^{onset}. \tag{1}$$

- **Wrong Rate-k** stands for the proportion of mistaken predictions when no event will happen in the future $k$ seconds. Formally,

$$Wr(k) = 1 - \frac{1}{|\mathcal{E}^P|} \sum_{t \in \mathcal{E}^P} max_{0 \leq s < k} y_{t+s}^{onset}. \tag{2}$$

In real-time clinical monitoring scenarios, a higher diagnosis rate indicates that the algorithm can better discover the onset of epileptic seizures and is less likely to miss any seizure event. A lower wrong rate implies that the algorithm is less prone to erroneous diagnoses, yielding more reliable results. These metrics provide a better quantification of the latency and performance of methods in the context of real-time detection, compared to common metrics such as precision and recall.

### 2.2  OUR METHOD: DYNAMIC SEIZURE NETWORK

#### 2.2.1  BLOCK-WISE SPECTRAL EMBEDDING

During the examination of EEG signals, some abnormal waves deviating from the normal electrical patterns in a healthy brain provide crucial insights into brain dysfunction and abnormalities. For example, spike-wave complexes, characterized by brief and high-amplitude waveforms, signify sudden

---

[1]We do not differentiate the EEG records from different patients in both tasks in this paper.

and abnormal electrical activity in the brain. These waves provide informative cues for both seizure detection and localization. Nevertheless, conventional spectral domain techniques like the Fourier transform, which assumes signal distribution invariance over time, may not effectively capture the dynamic nature of brain activities if directly applied to the raw EEG signals.

Based on the above considerations, we contend that the embedding process should extract spectral features while preserving temporal dependencies. Therefore, we employ a block-wise spectral embedding module to embed the input EEG signals:

$$\mathbf{X}_i^{(B)} = \{\mathbf{x}_t | i \times B \le t < (i+1) \times B\}, \tag{3}$$

$$\mathbf{F}_i^{(B)} = \mathcal{F}_{positive}(\mathcal{F}_{FT}(\mathbf{X}_i^{(B)})), \tag{4}$$

$$\mathbf{H}_i^{(B)} = \left( log \left| \mathbf{F}_i^{(B)} \right| \right) \mathbf{W}^{emb} + \mathbf{b}^{emb}, 0 \le i < T, \tag{5}$$

where $\mathcal{F}_{FT}(\cdot)$ is the Fourier transformation. Function $\mathcal{F}_{positive}(\cdot)$ selects the positive frequency components. $\mathbf{W}^{emb}$ and $\mathbf{b}^{emb}$ are trainable parameters. $B$ is the block length.

Then, we concatenate the embedded blocks of all channels as tensor $\mathbf{H} \in \mathbb{R}^{T \times C \times D}$. $T$ is the length of the embedded tensor, which equals to $\lfloor \frac{W}{B} \rfloor$ on the seizure detection task and $\lfloor \frac{S}{B} \rfloor$ on the real-time detection task. The spectral features of the input EEG sample can be encoded within the hidden dimension $D$, while the evolution of brain signals is retained within the temporal dimension $T$.

### 2.2.2 EVOLUTIONARY BRAIN STATE MODELING

The temporal patterns of the EEG signals can reflect the latent dynamics of brain regions, which are important for understanding brain activities and identifying epileptic seizures. We introduce an evolutionary brain state modeling module that employs an evolutionary memory bank $\mathbf{M}$ to capture the latent state of brain regions and uses the gated update mechanism (Cho et al., 2014) to update this latent state chronologically.

Furthermore, since recurrent networks suffer from forgetting issues when modeling long sequences (Hochreiter & Schmidhuber, 1997) and are not good at concentrating attention on discriminative patterns, we additionally leverage the attention mechanism to assist in the memory updating process:

$$\mathbf{M}_t = GRU(\mathbf{M}_{t-1}, \mathbf{H}_t \parallel Attn(\mathbf{H}_t, \mathbf{H}_{0:t}, \mathbf{H}_{0:t})), \tag{6}$$

where $Attn(\cdot)$ denotes self-attention (Vaswani et al., 2017) operation. The initial hidden state $\mathbf{M}_0$ is set to zeros. $\parallel$ refers to the concatenation operation.

**Efficient Recurrent Attention Network.** However, the attention mechanism requires linear computational demands and memory requirements at each timestamp, which may not be efficient in real-time scenarios. Consequently, we simplify the self-attention operation by introducing a **R**ecurrent **A**ttention **N**etwork (RAN). Formally, given query $\mathbf{Q}_t = \mathbf{H}_t \mathbf{W}_Q$, key $\mathbf{K}_t = \mathbf{H}_t \mathbf{W}_K$ and value $\mathbf{V}_t = \mathbf{H}_t \mathbf{W}_V$, we define:

$$\mathbf{S}_t = \mathbf{a} \odot \mathbf{S}_{t-1} + \mathbf{w}_p \odot exp(\mathbf{K}_t)^T \mathbf{V}_t \tag{7}$$

$$\mathbf{Z}_t = \mathbf{a} \odot \mathbf{Z}_{t-1} + \mathbf{w}_p \odot exp(\mathbf{K}_t)^T, \tag{8}$$

$$RAN(\mathbf{H_t}) = LayerNorm(\frac{exp(\mathbf{Q}_t)\mathbf{S}_t}{exp(\mathbf{Q}_t)\mathbf{Z}_t}), \tag{9}$$

where vectors $\mathbf{w}_p$ and $\mathbf{a}$ respectively denote the positional encoding and the time decay factor. $\mathbf{S}$ and $\mathbf{Z}$ are attention memory banks set to zeros at the beginning. $\odot$ is Hadamard product. Compared with the conventional self-attention mechanism, RAN has constant inference time and memory cost, regardless of the sequence length. The detailed deduction is demonstrated in Appendix A.9.

**Module Architecture.** In summary, by combining the RAN and RNN, the overall architecture of the proposed evolutionary brain state modeling module can be described as:

$$\mathbf{r}_t = \sigma \left( \gamma * RAN(\mathbf{H}_t) + f_{hr}(\mathbf{H}_t) + f_{mr}(\mathbf{M}_{t-1}) \right), \tag{10}$$

$$\mathbf{z}_t = \sigma \left( \gamma * RAN(\mathbf{H}_t) + f_{hz}(\mathbf{H}_t) + f_{mz}(\mathbf{M}_{t-1}) \right), \tag{11}$$

$$\mathbf{n}_t = \tanh \left( \gamma * RAN(\mathbf{H}_t) + f_{hn}(\mathbf{H}_t) + \mathbf{r}_t \odot f_{mn}(\mathbf{M}_{t-1}) \right), \tag{12}$$

$$\mathbf{M}_t = (\mathbf{1} - \mathbf{z}_t) \odot \mathbf{M}_{t-1} + \mathbf{z}_t \odot \mathbf{n}_t, \tag{13}$$

where $f_{hr}$, $f_{mr}$, $f_{hz}$, $f_{mz}$, $f_{hn}$ and $f_{mn}$ are linear mappings. $\sigma$ is the *sigmoid* function and $\gamma$ is a scalar hyper-parameter to balance the current information and historical information.

For detection task, we employ a trainable $cls$ token to extract discriminative patterns with attention mechanism and obtain a low-dimensional temporal embedding, i.e. $\mathbf{H}^{time} = Attn(cls, \{\mathbf{M}_t | 0 \leq t < T\}, \{\mathbf{M}_t | 0 \leq t < T\})$. For the real-time detection task, we directly utilize the recently updated memory $\mathbf{M}_t$ at current timestamp $t$ to represent the temporal embedding, i.e. $\mathbf{H}^{time} = \mathbf{M}_t$.

### 2.2.3 DYNAMIC BRAIN CORRELATION LEARNING

During epileptic seizures, the abnormal waves often propagate from one brain region to another, inducing aberrations in other regions. In this section, we aim to model such diffusion processes among brain regions. We first learn the underlying connectivities between brain regions and utilize a graph propagation component to model the interdependencies among them.

**Adaptive Brain Connectivity Learning.** Since the specific propagation routes have not been explored, we intend to adaptively learn them from the input data. Specifically, given the summarized temporal embedding matrix $\mathbf{H}^{time} \in \mathbb{R}^{C \times D}$, we use dot-product to measure the pattern similarity between different channels:

$$\tilde{\mathbf{A}} = Softmax \left( \frac{(\mathbf{H}^{time}\mathbf{W}_{dst}^{ACL} + \mathbf{E})^T (\mathbf{H}^{time}\mathbf{W}_{src}^{ACL} + \mathbf{E})}{\sqrt{D}} \right), \tag{14}$$

$$\bar{\mathbf{A}} = \tilde{\mathbf{A}} - diag(\tilde{\mathbf{A}}), \tag{15}$$

$$\mathbf{A} = \bar{\mathbf{A}}\bar{\mathbf{D}}, \tag{16}$$

where $\mathbf{W}_{dst}^{ACL}$ and $\mathbf{W}_{src}^{ACL}$ are trainable mapping weights. $\bar{\mathbf{D}}$ is the in-degree matrix of $\bar{\mathbf{A}}$. $\mathbf{E}$ is a trainable channel embedding parameter to learn the static correlation among brain regions.

**Propagation Modeling.** We further use the following diffusion process adapted from GraphSAGE (Hamilton et al., 2017) to explicitly model the propagation of abnormal waves:

$$\hat{\mathbf{H}}_h^P = \left[ \mathbf{A}\mathbf{H}_{h-1}^P || \mathbf{H}_{h-1}^P \right] \mathbf{W}_h^P + \mathbf{b}_h^P, \tag{17}$$

$$\mathbf{H}_h^P = LayerNorm(\hat{\mathbf{H}}_h^P), h = 1, 2, \cdots, H \tag{18}$$

where $\mathbf{W}^P$ and $\mathbf{b}^P$ are parameters. $H$ is the number of diffusion steps. $\mathbf{H}_0^P$ equals to $\mathbf{H}^{time}$. It is worth noting that on the real-time detection task, the learned brain connectivity matrix $\mathbf{A}$ and the propagation process only depend on the current timestamp rather than the entire history. Therefore, the dynamics of brain correlations can be learned with constant inference time and memory cost.

### 2.2.4 OUTPUT MODULE

Given embedding $\mathbf{H}^P \in \mathbb{R}^{C \times D}$ from the last diffusion step, we employ a simple output module to obtain predicted probabilities of epileptic seizures for both the seizure detection task and the real-time detection task. Formally,

$$\hat{y} = \sigma \left( f_{cls}(Flat(\phi(\mathbf{H}^C))) \right), \tag{19}$$

where $\phi(\cdot)$ stands for an activation function, e.g. *Tanh*. $f_{cls}(\cdot)$ is a multi-layer perceptron layer.

### 2.2.5 COMPLEXITY ANALYSIS

Table 1 provides a theoretical analysis of the time and space complexity of both baselines and our proposed method, when processing a single EEG slice on the real-time detection task.

In summary, the overall time complexity of our proposed method is $\mathcal{O}(SClogS + C^2L)$ and the overall space complexity is $\mathcal{O}(S + C + L)$ with respect to an input EEG slice $\mathbf{X}_t$. Specifically, in the spectral embedding module, we use the Fast Fourier Transform to extract the spectral information from slice $\mathbf{X}_t \in \mathbb{R}^{S \times C}$ with time complexity $\mathcal{O}(SClogS)$. In the evolutionary brain state modeling module, the historical information can be condensed into memory banks $\mathbf{M}$, $\mathbf{S}$ and $\mathbf{Z}$. Therefore it can be retrieved instantly without attending to the entire history, which reduces the time complexity from $\mathcal{O}(TCL)$ to $\mathcal{O}(CL)$. In the dynamic brain correlation learning module, the time cost of both learning the connectivities and propagating the information is $\mathcal{O}(C^2L)$, which is indispensable for methods modeling channel correlations with graphs such as SageFormer and DCRNN.

Table 1: Complexity analysis of baselines and DSN on real-time detection task. $S$, $T$ and $C$ respectively stand for the length of the slice, the number of slices, and the number of channels. $L$ denotes the number of model layers. Other parameters are regarded as constants.

| Method | Embedding | | Temporal | | Channel | |
|---|---|---|---|---|---|---|
| | Time | Space | Time | Space | Time | Space |
| DenseCNN | - | - | $\mathcal{O}(TSCL)$ | $\mathcal{O}(L)$ | $\mathcal{O}(C)$ | $\mathcal{O}(C)$ |
| TapNet | - | - | $\mathcal{O}(TSC)$ | $\mathcal{O}(TS)$ | $\mathcal{O}(TSCL)$ | $\mathcal{O}(CL)$ |
| STGCN | $\mathcal{O}(SClogS)$ | $\mathcal{O}(S)$ | $\mathcal{O}(TCL)$ | $\mathcal{O}(L)$ | $\mathcal{O}(C^2L)$ | $\mathcal{O}(L+C)$ |
| MTGNN | $\mathcal{O}(SClogS)$ | $\mathcal{O}(S)$ | $\mathcal{O}(TCL)$ | $\mathcal{O}(L)$ | $\mathcal{O}(C^2L)$ | $\mathcal{O}(L+C)$ |
| DCRNN-corr | $\mathcal{O}(SClogS)$ | $\mathcal{O}(S)$ | $\mathcal{O}(L)$ | $\mathcal{O}(L)$ | $\mathcal{O}(TC^2(D+L))$ | $\mathcal{O}(L)$ |
| TSD | $\mathcal{O}(SClogS)$ | $\mathcal{O}(SC+T)$ | $\mathcal{O}(T^2L)$ | $\mathcal{O}(L)$ | $\mathcal{O}(SC)$ | $\mathcal{O}(SC)$ |
| FEDFormer | $\mathcal{O}(SClogS)$ | $\mathcal{O}(S+T)$ | $\mathcal{O}(TClogT)$ | $\mathcal{O}(TL)$ | $\mathcal{O}(C)$ | $\mathcal{O}(C)$ |
| CrossFormer | $\mathcal{O}(SC)$ | $\mathcal{O}(S+TL)$ | $\mathcal{O}(T^2C)$ | $\mathcal{O}(L)$ | $\mathcal{O}(C)$ | $\mathcal{O}(L)$ |
| SageFormer | $\mathcal{O}(SClogS)$ | $\mathcal{O}(S+TC)$ | $\mathcal{O}(T^2CL)$ | $\mathcal{O}(L)$ | $\mathcal{O}(C^2L)$ | $\mathcal{O}(CL)$ |
| CNN-LSTM | $\mathcal{O}(SClogS)$ | $\mathcal{O}(S)$ | $\mathcal{O}(L)$ | $\mathcal{O}(L)$ | $\mathcal{O}(C)$ | $\mathcal{O}(C)$ |
| SegRNN | $\mathcal{O}(SClogS)$ | $\mathcal{O}(S)$ | $\mathcal{O}(L)$ | $\mathcal{O}(L)$ | $\mathcal{O}(C)$ | $\mathcal{O}(C)$ |
| LTransformer | $\mathcal{O}(SClogS)$ | $\mathcal{O}(S)$ | $\mathcal{O}(L)$ | $\mathcal{O}(L)$ | $\mathcal{O}(C)$ | $\mathcal{O}(C)$ |
| DCRNN-dist | $\mathcal{O}(SClogS)$ | $\mathcal{O}(S)$ | $\mathcal{O}(CL)$ | $\mathcal{O}(L)$ | $\mathcal{O}(C^2L)$ | $\mathcal{O}(L)$ |
| DSN | $\mathcal{O}(SClogS)$ | $\mathcal{O}(S)$ | $\mathcal{O}(CL)$ | $\mathcal{O}(L)$ | $\mathcal{O}(C^2L)$ | $\mathcal{O}(L+C)$ |

Some of the above methods, such as CNN-LSTM, SegRNN, LTransformer, DCRNN-dist and the proposed DSN, can process a single EEG slice with constant time and memory cost, regardless of the number of historical slices $T$. This characteristic enables them to detect epileptic seizures with reduced latency, making them more suitable for the task of real-time detection. The efficiency experiments can be found in Section 3.4.

## 3 EXPERIMENTS

### 3.1 EXPERIMENTAL SETUP

**Datasets.** We conduct our experiments on three real-world epilepsy datasets, including **FDUSZ**, **TUSZ** (Shah et al., 2018), and **CHBMIT** (Goldberger et al., 2000). The details and preprocessing procedure of the datasets are described in Appendix A.4.

**Baselines.** We compare our proposed approach with methods designed for multivariate time-series modeling, including **Shapelet** (Grabocka et al., 2014), **TapNet** (Zhang et al., 2020), **SegRNN** (Lin et al., 2023), **STGCN** (Yu et al., 2018), **MTGNN** (Wu et al., 2020), **FEDformer** (Zhou et al., 2022), **CrossFormer** (Zhang & Yan, 2023), **SageFormer** (Zhang et al., 2023), and **LTransformer** (Katharopoulos et al., 2020), and methods designed for epilepsy analysis, including **Dense-CNN** (Saab et al., 2020), **CNN-LSTM** (Ahmedt-Aristizabal et al., 2020), **DCRNN-dist**, **DCRNN-corr** (Tang et al., 2022) and **TSD** (Ma et al., 2023). We integrate the block-wise spectral embedding module to baselines except for Shapelet, TapNet, and DenseCNN to improve their performance. Appendix A.3 provides an introduction to these baselines.

**Metrics.** Following (Tang et al., 2022), we use $F_1$ and $AUC$ scores for seizure detection task, and also use $F_2$ score since missing any epileptic seizures is costly in clinical context (Chen et al., 2022). For the seizure real-time detection task, we use the diagnosis rate and the wrong rate with different $k$ to assess the models' capacity to identify latent seizure events and quantify the detection latency.

More detailed experimental setups are included in Appendix A.5.

### 3.2 PERFORMANCE ON THE SEIZURE DETECTION TASK

We evaluate the performance of all methods under two settings: transductive and inductive. In the transductive setting, the train/validation/test sets are partitioned chronologically for each subject, and all subjects are included in the training set. In the inductive setting, we split the train/validation/test sets across subjects, which emphasizes the ability to adapt the learned patterns to unseen patients. Table 2 shows the performance for all methods under the transductive and inductive settings.

From Table 2, we have the following conclusions: (1) *Learning the spectral information is essential for EEG representation learning.* Compared with methods that use raw EEG signals, such as Shapelet, DenseCNN, and TapNet, other methods generally exhibit superior performance. This

Table 2: Performance under the transductive and the inductive setting with window size 30-s. The best and the second-best results are boldfaced and underlined. The CHBMIT dataset is unavailable for the inductive setting due to the limited number of subjects.

| Method | Transductive | | | | | | | | | Inductive | | | | | |
| | FDUSZ | | | TUSZ | | | CHBMIT | | | FDUSZ | | | TUSZ | | |
| | $F_1$ | $F_2$ | $AUC$ | $F_1$ | $F_2$ | $AUC$ | $F_1$ | $F_2$ | $AUC$ | $F_1$ | $F_2$ | $AUC$ | $F_1$ | $F_2$ | $AUC$ |
|---|---|---|---|---|---|---|---|---|---|---|---|---|---|---|---|
| Shapelet | 0.313 | 0.393 | 0.734 | 0.313 | 0.405 | 0.822 | 0.068 | 0.085 | 0.602 | 0.319 | 0.501 | 0.704 | 0.291 | 0.373 | 0.738 |
| TapNet | 0.432 | 0.449 | 0.799 | 0.436 | 0.470 | 0.862 | 0.296 | 0.261 | 0.741 | 0.284 | 0.270 | 0.636 | 0.411 | 0.485 | 0.822 |
| DenseCNN | 0.544 | 0.553 | 0.846 | 0.585 | 0.581 | 0.928 | 0.268 | 0.254 | 0.826 | 0.490 | 0.485 | 0.750 | 0.561 | 0.622 | 0.877 |
| STGCN | 0.560 | 0.561 | 0.875 | 0.646 | 0.648 | 0.943 | 0.484 | 0.414 | 0.887 | 0.500 | 0.459 | 0.782 | 0.496 | 0.554 | 0.858 |
| MTGNN | 0.567 | 0.577 | 0.881 | 0.702 | 0.689 | 0.957 | 0.498 | 0.441 | 0.910 | 0.450 | 0.429 | 0.757 | 0.466 | 0.525 | 0.845 |
| SegRNN | 0.531 | 0.557 | 0.872 | 0.640 | 0.637 | 0.940 | 0.540 | 0.464 | 0.908 | 0.452 | 0.453 | 0.770 | 0.413 | 0.492 | 0.813 |
| CNN-LSTM | 0.602 | 0.593 | 0.898 | 0.666 | 0.664 | 0.947 | 0.582 | 0.546 | 0.909 | 0.497 | 0.478 | 0.789 | 0.451 | 0.519 | 0.843 |
| DCRNN-dist | 0.555 | 0.560 | 0.877 | 0.572 | 0.596 | 0.924 | 0.384 | 0.353 | 0.852 | 0.535 | 0.512 | 0.807 | 0.589 | 0.635 | 0.885 |
| DCRNN-corr | 0.608 | 0.605 | 0.895 | 0.604 | 0.602 | 0.934 | 0.472 | 0.443 | 0.894 | 0.556 | 0.523 | 0.811 | 0.597 | 0.613 | 0.890 |
| TSD | 0.547 | 0.555 | 0.882 | 0.500 | 0.536 | 0.909 | 0.494 | 0.473 | 0.902 | 0.474 | 0.462 | 0.767 | 0.421 | 0.500 | 0.831 |
| LTransformer | 0.543 | 0.567 | 0.888 | 0.542 | 0.553 | 0.920 | 0.269 | 0.280 | 0.833 | 0.481 | 0.494 | 0.792 | 0.444 | 0.552 | 0.855 |
| FEDFormer | 0.596 | 0.591 | 0.904 | 0.541 | 0.560 | 0.914 | 0.499 | 0.439 | 0.930 | 0.485 | 0.438 | 0.778 | 0.561 | 0.623 | 0.892 |
| CrossFormer | 0.578 | 0.574 | 0.898 | 0.713 | 0.722 | 0.964 | 0.525 | 0.464 | 0.925 | 0.476 | 0.461 | 0.791 | 0.408 | 0.480 | 0.813 |
| SageFormer | 0.593 | 0.597 | 0.906 | 0.711 | 0.704 | 0.964 | 0.576 | 0.538 | 0.935 | 0.477 | 0.475 | 0.794 | 0.415 | 0.493 | 0.815 |
| DSN | **0.633** | **0.620** | **0.915** | **0.739** | **0.727** | **0.968** | **0.611** | **0.570** | **0.949** | **0.567** | **0.545** | **0.841** | **0.613** | **0.644** | **0.900** |
| Improvements | 4.11% | 2.48% | 1.05% | 3.57% | 0.78% | 0.44% | 5.03% | 4.45% | 1.49% | 1.92% | 4.34% | 3.70% | 2.56% | 1.35% | 0.91% |

phenomenon suggests that explicitly incorporating the frequency component helps to distinguish abnormal brain waves and facilitates the learning of discriminative representations. (2) *The correlation between brain regions is significant for EEG representation learning.* SegRNN underperforms CNN-LSTM and DCRNN, primarily due to the lack of connectivity modeling. Similarly, TSD also underperforms CrossFormer and SageFormer, both of which incorporate connectivity modeling in their architecture. (3) *Capturing the dynamics of brain connectivity yields performance improvements.* By learning independent connectivities for different samples, DCRNN-corr outperforms its counterpart with static graphs, DCRNN-dist, in most cases. It demonstrates that the correlation of brain regions varies over time and cannot be fully exploited with a static graph. (4) *DSN outperforms existing methods that are designed for seizure detection and multivariate time-series modeling.* The superiority of our approach lies in the extraction of spectral information, the effective modeling of the evolutionary brain states, and its capability to capture dynamic brain connectivities.

## 3.3 PERFORMANCE ON THE REAL-TIME DETECTION TASK

We further evaluate the performance of the real-time detection task, which is more feasible in real-world clinical scenarios. For streaming methods (i.e. CNN-LSTM, GRU, DCRNN-dist, LTransformer, and DSN), the input at each timestamp $t$ is a single slice $\mathbf{X}_t \in \mathbb{R}^{S \times C}$. Slices of different timestamps are input sequentially. For other methods, we use a historical 15-s EEG clip as the input for each timestamp, i.e. $\mathbf{X}_{t-15:t} \in \mathbb{R}^{15S \times C}$, at timestamp $t$. The performance of baselines with different horizons $k$ are shown in Figure 3.

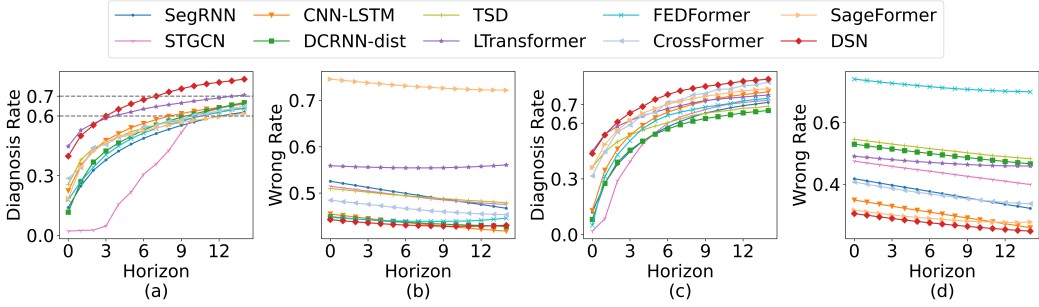

Figure 3: Performance of seizure real-time detection with different horizons. **(a)-(b)** Performance on FDUSZ dataset. **(c)-(d)** Performance on TUSZ dataset.

Among all methods, our proposed DSN (red lines) has the highest diagnosis rate and lowest wrong rate in most cases. Our method is able to detect over 60% of seizure events within 5 seconds and approximately 80% of seizure events within 15 seconds after the onset of seizures. More specifically, the diagnosis rates within 5/10/15 seconds are 0.690/0.798/0.838 on the TUSZ dataset and 0.635/0.737/0.786 on the FDUSZ dataset, respectively 4.2% and 8.9% higher than the best baseline

on average. The wrong rates within 5/10/15 seconds are 0.289/0.277/0.272 on the TUSZ dataset and 0.422/0.407/0.389 on the FDUSZ dataset, respectively 8.2% and 0.7% better than the best baseline on average. In other words, DSN is capable of detecting seizure events with lower latency. For example, DSN achieves diagnosis rates of 60% and 70% respectively 1 second and 7 seconds faster than the best baseline, LTransformer, on the FDUSZ dataset (see the gray horizontal lines in Figure 3 (a)). This capability to swiftly identify seizure events within a shorter latency is attributed to the recurrent attention mechanism, which attentively detects abnormal brain wave patterns, and the dynamic brain correlation learning module to capture the propagation of abnormal signals.

## 3.4 EFFICIENCY COMPARISON

We demonstrate the efficiency of DSN by comparing the inference time and GPU memory usage for both the detection task and the real-time detection task. We report the average results over 20 epochs on the TUSZ dataset in Figure 4. The GPU usages are obtained by `nvidia-smi` command.

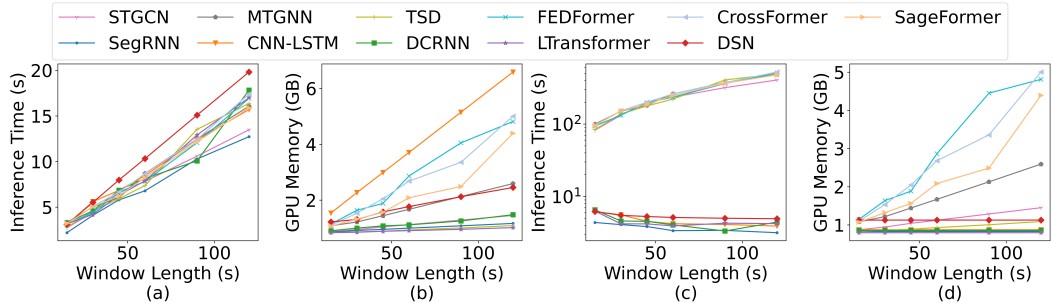

Figure 4: **(a)-(b)** Inference time and GPU usage on seizure detection task. **(c)-(d)** Inference time and GPU usage on real-time detection task. All results are obtained with batch size 256. DenseCNN and DCRNN-corr are excluded because their inference time is too long.

Concretely, the inference time of DSN in the detection task is 3.1/5.5/8.0/10.3/15.1/19.8 seconds when the window length is 15/30/45/60/90/120 seconds, which is slightly longer than baselines (Figure 4 (a)), primarily because of the overhead of constructing dynamic graphs for every timestamp. However, unlike CrossFormer and SageFormer, whose memory cost grows quadratically with the sequence length, the memory cost of DSN scales linearly with the window length (Figure 4 (b)), due to the effective design of recurrent attention mechanism.

On the real-time detection task, where the EEG slices are input sequentially, DSN and other streaming methods are able to retrieve historical information directly from memory banks. Consequently, they output seizure probabilities with shorter inference times (Figure 4 (c)) and maintain nearly constant memory costs (Figure 4 (d)). As a result, these approaches are particularly advantageous in real-world seizure monitoring scenarios. Further details of detection performance with respect to window length can be found in Appendix A.7.

## 3.5 ABLATION STUDY

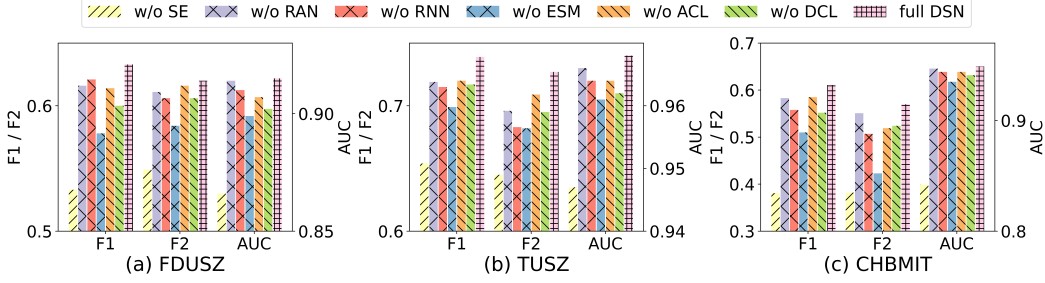

Figure 5: Effects of different components on all the datasets.

In this section, we perform an ablation study to verify the effectiveness of the proposed modules by removing each component and examining their impact on seizure detection performance: (1) replacing *block-wise spectral embedding* module with a fully-connected layer (w/o SE), (2) removing *recurrent attention networks* (w/o RAN), (3) removing *recurrent neural networks* (w/o RNN), (4)

removing the entire *evolutionary state modeling* module (w/o ESM), (5) replacing the *adaptive brain connectivity learning* component with static graph learning component proposed in Wu et al. (2020) (w/o ACL) and (6) removing the entire *dynamic brain correlation learning* module (w/o DCL).

From Figure 5, we observe that DSN achieves the best performance with all components, and removing any individual component will lead to a worse result. In particular, SE explicitly extracts the spectral information from raw EEG signals. ESM is composed of RAN and RNN, which respectively localize the abnormal brain activities and capture the evolution of brain region states. ACL learns dynamic brain interconnections and DCL models the propagation of abnormal waves among brain regions. Therefore, the effectiveness of each component is verified.

## 3.6 VISUALIZATION

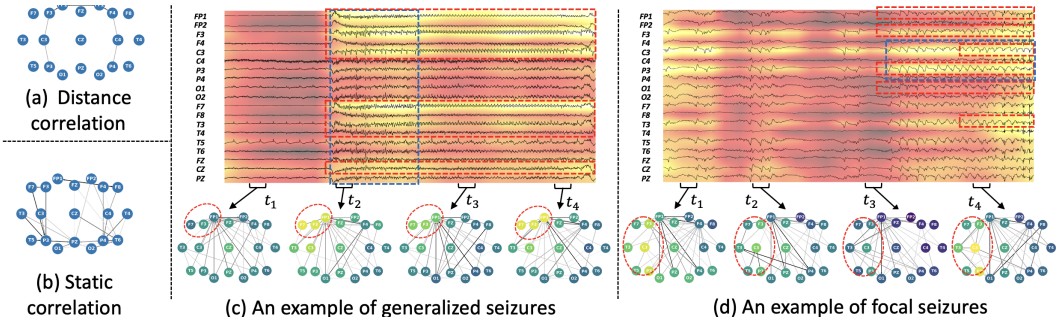

Figure 6: Visualization of the predefined **(a)**, static **(b)** and dynamic **(c)-(d)** correlation graphs and the saliency maps for two random samples. Brighter colors indicate higher saliency and darker lines represent stronger connections. Red circles refer to possible onset regions. Red boxes refer to annotated seizure labels. More samples can be found in Appendix A.10.

To illustrate the interpretability of DSN, we generate visualizations of the learned brain correlations and overlay the attention weights $\mathbf{Z}$ (see Equation 7) onto the raw EEG clips to reveal the hotspots identified by our approach. In Figure 6, we observe that areas of high saliency are concentrated within abnormal brain wave patterns, which might provide assistance in the diagnosis of seizure onsets and the localization of abnormal brain activities. For instance, in Figure 6 (c), the highlighted spike waves (blue rectangle) that started at $t_2$ signify the onset event of a generalized seizure. Similarly, in Figure 6 (d), the highlighted sharp waves (blue rectangle) observed in electrodes C3 and P3 indicate the abnormal brain region might be the left temporal lobe. We further use red boxes to indicate the annotated labels of seizure events, where the hotspots overlap well with the ground truth labels. The localization capability of the recurrent attention mechanism benefits the swift identification of abnormal brain activities and facilitates low-latency seizure real-time detection.

Furthermore, compared with the distance-based brain region graph (Figure 6 (a), Tang et al. (2022)) or the learned static graph (Figure 6 (b)), the dynamic brain connectivities can better illustrate the evolution patterns of brain states and region correlations. In Figure 6 (d), we observe that the state of the left temporal lobe (red circle) is active at $t_1$, turns stationary from $t_2$ to $t_3$, and then becomes active again at $t_4$. This pattern corresponds to the burst-suppression patterns commonly associated with epileptic seizures. However, it is worth noting that our visualization results require rigorous review by clinical experts in real-world clinical settings.

## 4 CONCLUSION

In this paper, we aimed to investigate a framework for both seizure detection and real-time detection tasks, which identifies both the presence and the latency of epileptic seizure onsets under real-time context. We presented two metrics to quantify the latency and correctness of seizure detection in real-world monitoring scenarios. Moreover, in order to effectively learn dynamic electroencephalography representations, we designed a framework DSN consisting of a block-wise spectral embedding module, an evolutionary brain state modeling module, and a dynamic brain correlation learning module to capture the discriminative spectral patterns and the dynamics of brain states and region correlations from EEG records. Theoretical analysis and experimental results on three real-world clinical datasets demonstrated the effectiveness and efficiency of the proposed method. The visualization study further illustrated its interpretability. Our work encouraged future research to rethink the latency and efficiency of automated seizure detection methods in real-world clinical scenarios.

## ETHICS STATEMENT

The FDUSZ, TUSZ and CHBMIT datasets used in this study have been de-identified and anonymized. Any personally identifiable information has been removed. The seizure detection models described in this study do not yield any harmful insights. We strongly insist that any insights derived from this study be subject to rigorous validation by board-certified neurologists or relevant experts in real-world clinical settings.

## REPRODUCIBILITY STATEMENT

The code of DSN can be found in the supplementary material. The TUSZ and CHBMIT datasets are publicly available. The preprocessing steps are detailed in Appendix A.4 and the preprocessing code can be found in the supplementary material.

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

# A  APPENDIX

## A.1  RELATED WORKS

In this section, we review the related works and show the differences between previous works and our method, including methods designed for multivariate time-series modeling and seizure analysis.

### A.1.1  MULTIVARIATE TIME-SERIES REPRESENTATION LEARNING

Due to the prevalence of multivariate time series in real-world scenarios, numerous efforts have been made on multivariate time series modeling, including classification (Schäfer & Leser, 2018; Zheng et al., 2014; Karim et al., 2019; Zhang et al., 2020), forecasting (Yu et al., 2018; Wu et al., 2020; Zhou et al., 2022; Zhang & Yan, 2023), and anomaly detection (Zhang et al., 2019; Audibert et al., 2020; Zhao et al., 2020). For multivariate time series classification, Grabocka et al. (2014); Karlsson et al. (2016) extracted representative patterns named shapelets as features to discriminate time series classes. Karim et al. (2019) learned low-dimensional representations with deep learning methods. TapNet (Zhang et al., 2020) proposed an attentional prototype network to train the feature representation based on their distance to class prototypes. SegRNN (Lin et al., 2023) modeled long-term series with block-wise RNNs (Cho et al., 2014). For multivariate time series forecasting, DCRNN (Li et al., 2018) combined RNNs and GCNs (Kipf & Welling, 2017) to jointly model the evolving hidden states and channel correlations of multivariate time series. STGCN (Yu et al., 2018) applied convolutional structures on both channel and temporal dimensions to extract spatial-temporal representations of multivariate time series. Wu et al. (2020) further exploit inherent channel relationships with adaptively learned graph structures. Moreover, some recent works employed Transformer (Vaswani et al., 2017) models as backbones. Autoformer (Wu et al., 2021) and FEDformer (Zhou et al., 2022) introduced decomposition methods for long-term time series forecasting task. Crossformer (Zhang & Yan, 2023) and SageFormer (Zhang et al., 2023) respectively enhanced cross-channel dependency learning with attention mechanism and GNNs.

### A.1.2  EPILEPTIC SEIZURE ANALYSIS FROM EEG SIGNALS

Electroencephalography (EEG) signals are effective diagnostic tools for studying brain activities during epileptic seizures. Recent developments in epileptic seizure analysis benefit from the advancement of data analysis methods and approaches based on deep learning. Saab et al. (2020) proposed Dense-CNN with pure convolutional networks to detect epileptic seizures from raw EEG signals. CNN-LSTM (Ahmedt-Aristizabal et al., 2020) enhanced convolutional networks with external memory modules for seizure classification. Tang et al. (2022) represented the non-Euclidean spatial dependencies in EEG signals with graphs and constructed two graph structures that respectively capture the electrode geometry and dynamic brain connectivity. Their proposed self-supervised method improved the performance of both seizure detection and classification. BrainNet (Chen et al., 2022) modeled the propagation process of epileptic waves with a graph diffusion mechanism. Peng et al. (2022) enhanced the convolution layer of EEGNet (Lawhern et al., 2018) with sinusoidal encoding module. TSD (Ma et al., 2023) proposed a Transformer-based method to detect epileptic seizures in the frequency domain of EEG signals. Albaqami et al. (2023) used a CNN module and a Bi-LSTM module with attention to learn two different EEG representations for seizure classification. Tang et al. (2023) leveraged the Structured State Spaces architecture (Gu et al., 2021) to capture long-range temporal dependencies and learned dynamically evolving graph structures in raw EEG signals. DeepSOZ (Shama et al., 2023) utilized attention and bi-LSTMs for precise seizure detection in each timestamp. Compared with the above methods, our DSN are more suitable for streaming EEG inputs with better performance and higher efficiency.

Although insightful, most of these methods are only applicable to fixed-length input samples and therefore have higher latency and resource utilization when the inputs are streaming. Moreover, they either lack the ability to localize abnormal brain waves or capture the evolution of brain states and the dynamics of brain correlations, which limits their performance on EEG representation learning.

## A.2 ALGORITHM

The pseudocode of DSN on seizure detection task and on real-time detection task are respectively described in Algorithm 1 and Algorithm 2.

---

**Algorithm 1:** Pseudocode of DSN on seizure detection task.

---

**Input**   : Input EEG clip $\mathbf{X}$ with length $T$
**Output:** Predicted seizure probability $\hat{y}$

```
/* Initial memory banks                                        */
```
$\mathbf{S}_0, \mathbf{Z}_0, \mathbf{M}_0 \leftarrow \mathbf{0}, \mathbf{0}, \mathbf{0};$
```
/* Spectral Embedding                                          */
```
**for** $t \in 1, 2, \cdots, T$ **do**
  $\mathbf{H}_t \leftarrow SpectralEmbedding(\mathbf{X}_t);$
**end**
```
/* Evolutionary Brain State Modeling                           */
```
**for** $t \in 1, 2, \cdots, T$ **do**
  $\mathbf{Q}_t, \mathbf{K}_t, \mathbf{V}_t \leftarrow \mathbf{H}_t\mathbf{W}_Q, \mathbf{H}_t\mathbf{W}_K, \mathbf{H}_t\mathbf{W}_V;$
  $\mathbf{S}_t, \mathbf{Z}_t \leftarrow UpdateRAN(\mathbf{S}_{t-1}, \mathbf{Z}_{t-1}, \mathbf{K}_t, \mathbf{V}_t);$
  $\mathbf{O}_t \leftarrow OutputRAN(\mathbf{Q}_t, \mathbf{S}_t, \mathbf{Z}_t);$
  $\mathbf{M}_t \leftarrow RNN(\mathbf{O}_t, \mathbf{H}_t, \mathbf{M}_{t-1});$
**end**
$\mathbf{H}^{time} \leftarrow Attention(cls, \{\mathbf{M}_t\}, \{\mathbf{M}_t\});$
```
/* Dynamic Brain Correlation Learning                          */
```
$\mathbf{A} \leftarrow ConnectivityLearning(\mathbf{H}^{time});$
$\mathbf{H}^P \leftarrow Propagation(\mathbf{H}^{time}, \mathbf{A});$
```
/* Output                                                      */
```
$\hat{y} \leftarrow Output(\mathbf{H}^P)$

---

**Algorithm 2:** Pseudocode of DSN on real-time detection task.

---

**Input**   : Input EEG slice $\mathbf{X}_t$, previous RAN memory $\mathbf{S}_{t-1}$ and $\mathbf{Z}_{t-1}$, previous RNN memory $\mathbf{M}_{t-1}$. All memory is set to zero when $t = 0$.
**Output:** Predicted seizure probability $\hat{y}_t$, updated RAN memory $\mathbf{S}_t$ and $\mathbf{Z}_t$, updated RNN memory $\mathbf{M}_t$

```
/* Spectral Embedding                                          */
```
$\mathbf{H}_t \leftarrow SpectralEmbedding(\mathbf{X}_t);$
```
/* Evolutionary Brain State Modeling                           */
```
$\mathbf{Q}_t, \mathbf{K}_t, \mathbf{V}_t \leftarrow \mathbf{H}_t\mathbf{W}_Q, \mathbf{H}_t\mathbf{W}_K, \mathbf{H}_t\mathbf{W}_V;$
$\mathbf{S}_t, \mathbf{Z}_t \leftarrow UpdateRAN(\mathbf{S}_{t-1}, \mathbf{Z}_{t-1}, \mathbf{K}_t, \mathbf{V}_t);$
$\mathbf{O}_t \leftarrow OutputRAN(\mathbf{Q}_t, \mathbf{S}_t, \mathbf{Z}_t);$
$\mathbf{M}_t \leftarrow RNN(\mathbf{O}_t, \mathbf{H}_t, \mathbf{M}_{t-1});$
$\mathbf{H}^{time} \leftarrow \mathbf{M}_t;$
```
/* Dynamic Brain Correlation Learning                          */
```
$\mathbf{A} \leftarrow ConnectivityLearning(\mathbf{H}^{time});$
$\mathbf{H}^P \leftarrow Propagation(\mathbf{H}^{time}, \mathbf{A});$
```
/* Output                                                      */
```
$\hat{y}_t \leftarrow Output(\mathbf{H}^P)$

---

## A.3 BASELINES

We compare our proposed approach with multivariate time-series modeling methods, including methods for classification and forecasting:

- Shapelet (Grabocka et al., 2014) extracted discriminative sub-sequences from time series for classification.

- TapNet (Zhang et al., 2020) combines deep learning methods and traditional methods with attentional networks for time series classification.
- STGCN (Yu et al., 2018) applied convolutional structures for spatial and temporal representation learning.
- MTGNN (Wu et al., 2020) proposed adaptively learned graph structures to exploit inherent spatial relationships.
- SegRNN (Lin et al., 2023) modeled long-term time series with block-wise recurrent neural networks.
- LTransformer (Katharopoulos et al., 2020) proposed an autoregressive Transformer with an efficient linear attention mechanism.
- FEDformer (Zhou et al., 2022) combined Transformer with the seasonal-trend decomposition and proposed a frequency-enhanced attention mechanism.
- Crossformer (Zhang & Yan, 2023) respectively modeled temporal and spatial correlation with two individual attention modules.
- SageFormer (Zhang et al., 2023) introduced a graph-enhanced Transformer model for multivariate time-series forecasting.

For methods designed for forecasting tasks (e.g. MTGNN and FEDformer), we only use their encoders to obtain the learned low-dimension representations. We also compare our approach with methods specifically proposed for epilepsy analysis on EEG signals:

- Dense-CNN (Saab et al., 2020) used pure convolutional networks for automated EEG feature extraction.
- CNN-LSTM (Ahmedt-Aristizabal et al., 2020) combined convolutional networks and recurrent networks for seizure classification. We use the last hidden state of the LSTM component for the detection task and all hidden states for the real-time detection task.
- DCRNN-dist (Tang et al., 2022) constructed graphs with electrode geometry to model the spatial dependencies of EEG signals and leveraged recurrent networks for temporal modeling.
- DCRNN-corr (Tang et al., 2022) learned the electrode graph from input EEG signals for dynamic brain connectivity modeling.
- TSD (Ma et al., 2023) utilized a self-attention mechanism to detect seizure events from EEG signals.

Since the open-source code for BrainNet is currently available and we failed to reproduce the reported performance of GraphS4mer in the original paper, we decide not to use them as our baselines.

For all baselines as well as DSN, we use identical output modules for a fair comparison. Moreover, we introduce the proposed block-wise spectral embedding component (see Section 2.2.1) into the following baselines: SegRNN, STGCN, MTGNN, FEDformer, CrossFormer, SageFormer, CNN-LSTM, and DCRNN and use the embedded spectral embedding $\mathbf{H}$ as their input to improve the performance of seizure detection.

A.4 DATASET DESCRIPTION

We conducted our experiments on three epilepsy datasets:

- **FDUSZ** is a large-scale anonymous seizure dataset containing 260 seizure records and 51 healthy records. This dataset contains 1,598 hours EEG records in total including 152 hours seizure records. The sample rate varies from 500Hz to 2,000Hz. We regard each record file as an individual subject. 12 common channels are selected, including an electrocardiogram (ECG) channel and an electromyography (EMG) channel.
- **TUSZ** (TUH EEG Seizure Corpus[2], Shah et al. (2018)) is a large-scale seizure dataset derived from The Temple University Hospital EEG Data Corpus. We use the latest version

---

[2]https://isip.piconepress.com/projects/tuh_eeg/

v2.0.0, which contains more than 1,476 hours of EEG records from 675 anonymous subjects, including 76 hours of seizure records from 287 patients. The sample rate ranges from 250Hz to 1,000Hz. We select 19 common electrodes that are included in all record files.

- **CHBMIT** (CHB-MIT Scalp EEG Database[3], Goldberger et al. (2000) is a collection of EEG recordings of 23 anonymous pediatric cases from the Children's Hospital Boston. This dataset contains a total of 664 .edf files including 198 seizure events. All signals are sampled at 256Hz. We select 18 electrodes that are recorded in all files and discard files that have no seizures.

The EEG electrodes in the datasets follow the standard 10-20 EEG electrode placement (Jasper, 1958). The selected electrodes in the datasets are listed in Table 3. All seizure records are labeled by epileptologists to indicate the start timestamp and the end timestamp of each epilepsy event, which can be used as ground truths for both tasks.

Table 3: Selected channels of each dataset.

| Dataset | #Channels | Channels |
|---------|-----------|----------|
| FDUSZ | 12 | *Fp1, Fp2, F3, F4, C3, C4, P3, P4, T3, T4, EKG, EMG* |
| TUSZ | 19 | *Fp1, Fp2, F3, F4, C3, C4, P3, P4, O1, O2, F7, F8, T3, T4, T5, T6, FZ, CZ, PZ* |
| CHBMIT | 18 | *Fp1-F7, F7-T7, T7-P7, P7-O1, Fp1-F3, F3-C3, C3-P3, P3-O1, Fp2-F4, F4-C4, C4-P4, P4-O2, FP2-F8, F8-T8, T8-P8, P8-O2, FZ-CZ, CZ-PZ* |

**Clipping samples**. To align the sample rate of different records, we resample all EEG signals down to 100Hz using the `resample` function in SciPy package (Virtanen & Gommers, 2020), in accordance with the typical frequency band of brain activities (Rasheed et al. (2021)). For the seizure detection task, we clip the EEG records by sliding a 30-second window without overlaps and drop the last window if it is shorter than 30 seconds. The label of each EEG clip is 1 if at least one seizure event happens within that clip, otherwise, the label is 0. Therefore, each input EEG clip can be denoted by $\mathbf{X} \in \mathbb{R}^{T \times C}$, where $T = 3000$ is the length of the clip and $C$ is the number of electrodes, and its binary label can be denoted by $y^{det} \in \{0, 1\}$ to indicate whether a seizure event occurs within the clip or not. For the real-time detection task, we slice the EEG records by sliding a 1-second window without overlaps and obtain EEG slices and the corresponding output $\{(\mathbf{X}_t, y_t^{onset}) | t = 0, 1, 2, \cdots\}$, where $\mathbf{X}_t \in \mathbb{R}^{S \times C}$ is the slice at timestamp $t$ with $y_t^{onset} \in \{0, 1\}$ as its label. $S = 100$ is the slice length. These slices will be fed into the models sequentially in a streaming manner. In this paper, we obtain the slices from EEG clips rather than the raw records for parallel training.

**Splitting datasets.** We evaluate the methods under two different settings, transductive and inductive. Table 4 shows the statistics of the datasets. Under the transductive setting, all subjects are included in the training set. We arrange the clips and slices of each subject in chronological order and select the first 70% clips/slices of each subject as the training set, 10% as the validating set, and the last 20% as the testing set. Under the inductive setting, the training set, validating set, and testing set have distinct subjects. On FDUSZ datasets, we randomly select 70% subjects (including patients and healthy subjects) as the training set, 10% as the validation set, and 20% as the testing set. On TUSZ dataset, since the train/validation/test sets are already split by subjects, we directly use them for the inductive seizure detection task. The CHBMIT dataset is not available for inductive settings since the validation set only contains 2 subjects.

Since the datasets are imbalanced, we randomly under-sample the negative samples in the training set such that there are equal positive and negative samples in the training set. We use all samples in the validation set and the test set.

### A.5 IMPLEMENTATION DETAILS

All methods are trained for 100 epochs with batch size 256 and early stop when the validation loss does not decrease for 20 consecutive epochs. The average results from 5 runs with different random seeds are reported. For both the seizure detection task and the seizure real-time detection task, we use Binary Cross Entropy as the loss function with Adam (Kingma & Ba, 2017) optimizer and cosine

---

[3]https://physionet.org/content/chbmit/1.0.0/

Table 4: Statistics of the datasets.

| Setting | Dataset | Train | | Validate | | Test | |
|---------|---------|-------|---|----------|---|------|---|
| | | Clips (% Seizure) | Patients (% Seizure) | Clips (% Seizure) | Patients (% Seizure) | Clips (% Seizure) | Patients (% Seizure) |
| Transductive | FDUSZ | 133769 (12.1%) | 311 (75.2%) | 18985 (11.6%) | 311 (44.1%) | 38540 (9.8%) | 311 (53.1%) |
| | TUSZ | 95377 (8.1%) | 674 (40.4%) | 9604 (10.0%) | 663 (22.2%) | 35392 (6.0%) | 674 (26.9%) |
| | CHBMIT | 15273 (2.5%) | 24 (100.0%) | 2168 (1.5%) | 24 (45.8%) | 4396 (2.5%) | 24 (91.7%) |
| Inductive | FDUSZ | 130274 (10.2%) | 217 (79.2%) | 23347 (14.6%) | 31 (77.4%) | 37673 (14.7%) | 63 (77.8%) |
| | TUSZ | 98732 (7.2%) | 578 (34.9%) | 27357 (8.1%) | 53 (84.9%) | 14284 (8.7%) | 43 (79.1%) |

annealing learning rate scheduler (Loshchilov & Hutter, 2016). All the experiments are conducted on a Ubuntu machine equipped with Intel(R) Xeon(R) Gold 6130 and NVIDIA Tesla T4 with 16GB memory, with Python 3.10 and PyTorch 1.12 (Paszke et al., 2019).

The hyper-parameters of Shapelet, Dense-CNN, CNNLSTM, DCRNN-dist and DCRNN-corr are largely adopted from Grabocka et al. (2014) and Tang et al. (2022). We set the hidden dimension of other methods to identical 64 for a fair comparison, following Yu et al. (2023). For other hyperparameters such as the number of layers, learning rate and dropout rate, in addition to following their official settings, we perform an exhaustive grid-search to find the optimal configurations independently for all methods, for all datasets and for both tasks. For our proposed method, the hyper-parameters used are listed in Table 5. The length of blocks $B$ is set to 1-second, following (Tang et al., 2022), which equals to 100. The number of diffusion steps in Equation 17 is 1.

Table 5: Best hyper-parameters of DSN on different datasets.

| Hyper-parameter | Transductive | | | Inductive | | Real-time | |
|-----------------|--------------|------|--------|-----------|------|-----------|------|
| | FDUSZ | TUSZ | CHBMIT | FDUSZ | TUSZ | FDUSZ | TUSZ |
| $\gamma$ | 1 | 1 | 0.05 | 1 | 1 | 1 | 1 |
| #Layer of ESM | 1 | 1 | 2 | 2 | 1 | 1 | 1 |
| #Layer of DCL | 1 | 1 | 1 | 1 | 1 | 1 | 1 |
| Activation | Tanh | Tanh | Relu | Tanh | Tanh | Tanh | Tanh |
| Classifier | FC | FC | FC | FC | Max | FC | FC |
| Dropout rate | 0.0 | 0.0 | 0.0 | 0.0 | 0.0 | 0.0 | 0.0 |

To obtain seizure/non-seizure predictions, we perform a decision threshold search on the validation set and select the decision threshold with the highest $F_1$ score, following (Tang et al., 2022). EEG clips/slices with probabilities higher than the threshold will be regarded as seizures, while clips/slices with probabilities lower than the threshold will be regarded as non-seizures. We further investigate the sensitivity of the threshold in Appendix A.8.

## A.6 SELF-SUPERVISED PRETRAINING

Several previous works (Tang et al., 2022; Das et al., 2022) emphasized the effectiveness of self-supervised pretraining (SSL) on EEG signals to facilitate downstream tasks such as seizure detection. We also conducted experiments to further evaluate the importance of SSL. We pretrained the model to predict the next 30-s preprocessed EEG clip given a preprocessed 30-s EEG clip and kept other settings identical to the original paper (Tang et al., 2022), where the early-stop patience was 5 and the learning rate was 0.0001. Moreover, we used another setting with patience 20 and learning rate 0.001. The comparison between methods with or without SSL is shown in Table 6.

Different from Tang et al. (2022), we observe that self-supervised pretraining does little help to the performance of seizure detection. We owe this phenomenon to that the setting adopted in (Tang et al. (2022), with patience 5 and learning rate 0.0001) is not the optimal setting for either FDUSZ or TUSZ datasets and will lead to under-fitting issues if training the model from sketch with detection labels. To this end, we believe that with a proper setting (i.e. patience 5 and learning rate 0.001),

models trained from sketch can achieve comparable performance and decide not to use the self-supervised pretraining strategy in this paper.

Table 6: Performance of methods with/without self-supervised pretraining on FDUSZ-Transductive and TUSZ-Transductive datasets.

| Dataset | Method | Setting | | | $Acc.$ | $Prec.$ | $Recall$ | $F_1$ | $F_2$ | $AUC$ |
|---|---|---|---|---|---|---|---|---|---|---|
| | | Patience | Learning Rate | Pretrain | | | | | | |
| FDUSZ | SegRNN | 5 | 1e-4 | × | 0.891 | 0.451 | 0.512 | 0.479 | 0.498 | 0.845 |
| | | 5 | 1e-4 | ✓ | 0.896 | 0.480 | 0.589 | 0.526 | 0.561 | 0.873 |
| | | 20 | 1e-3 | × | 0.901 | 0.494 | 0.576 | 0.531 | 0.557 | 0.872 |
| | | 20 | 1e-3 | ✓ | 0.901 | 0.498 | 0.580 | 0.534 | 0.560 | 0.869 |
| | DCRNN-dist | 5 | 1e-4 | × | 0.897 | 0.484 | 0.513 | 0.496 | 0.506 | 0.838 |
| | | 5 | 1e-4 | ✓ | 0.913 | 0.558 | 0.563 | 0.559 | 0.561 | 0.871 |
| | | 20 | 1e-3 | × | 0.912 | 0.547 | 0.563 | 0.555 | 0.560 | 0.877 |
| | | 20 | 1e-3 | ✓ | 0.913 | 0.556 | 0.565 | 0.559 | 0.562 | 0.870 |
| | TSD | 5 | 1e-4 | × | 0.879 | 0.413 | 0.561 | 0.475 | 0.523 | 0.848 |
| | | 5 | 1e-4 | ✓ | 0.905 | 0.511 | 0.543 | 0.527 | 0.537 | 0.870 |
| | | 20 | 1e-3 | × | 0.909 | 0.534 | 0.561 | 0.547 | 0.555 | 0.882 |
| | | 20 | 1e-3 | ✓ | 0.906 | 0.515 | 0.540 | 0.527 | 0.535 | 0.869 |
| | CrossFormer | 5 | 1e-4 | × | 0.897 | 0.478 | 0.544 | 0.507 | 0.528 | 0.868 |
| | | 5 | 1e-4 | ✓ | 0.918 | 0.580 | 0.593 | 0.586 | 0.590 | 0.902 |
| | | 20 | 1e-3 | × | 0.919 | 0.585 | 0.572 | 0.578 | 0.574 | 0.898 |
| | | 20 | 1e-3 | ✓ | 0.917 | 0.574 | 0.597 | 0.584 | 0.592 | 0.900 |
| TUSZ | SegRNN | 5 | 1e-4 | × | 0.940 | 0.511 | 0.567 | 0.536 | 0.554 | 0.914 |
| | | 5 | 1e-4 | ✓ | 0.958 | 0.650 | 0.649 | 0.650 | 0.649 | 0.945 |
| | | 20 | 1e-3 | × | 0.957 | 0.644 | 0.636 | 0.640 | 0.637 | 0.940 |
| | | 20 | 1e-3 | ✓ | 0.956 | 0.627 | 0.676 | 0.650 | 0.665 | 0.944 |
| | DCRNN-dist | 5 | 1e-4 | × | 0.931 | 0.445 | 0.536 | 0.486 | 0.515 | 0.891 |
| | | 5 | 1e-4 | ✓ | 0.946 | 0.549 | 0.611 | 0.578 | 0.597 | 0.923 |
| | | 20 | 1e-3 | × | 0.944 | 0.540 | 0.614 | 0.572 | 0.596 | 0.924 |
| | | 20 | 1e-3 | ✓ | 0.947 | 0.561 | 0.593 | 0.576 | 0.586 | 0.922 |
| | TSD | 5 | 1e-4 | × | 0.928 | 0.420 | 0.518 | 0.463 | 0.494 | 0.889 |
| | | 5 | 1e-4 | ✓ | 0.933 | 0.458 | 0.551 | 0.499 | 0.528 | 0.904 |
| | | 20 | 1e-3 | × | 0.932 | 0.451 | 0.563 | 0.500 | 0.536 | 0.909 |
| | | 20 | 1e-3 | ✓ | 0.934 | 0.464 | 0.557 | 0.505 | 0.534 | 0.906 |
| | CrossFormer | 5 | 1e-4 | × | 0.941 | 0.513 | 0.513 | 0.511 | 0.512 | 0.908 |
| | | 5 | 1e-4 | ✓ | 0.962 | 0.690 | 0.672 | 0.680 | 0.675 | 0.957 |
| | | 20 | 1e-3 | × | 0.965 | 0.701 | 0.727 | 0.713 | 0.722 | 0.964 |
| | | 20 | 1e-3 | ✓ | 0.966 | 0.727 | 0.692 | 0.708 | 0.698 | 0.960 |

## A.7 SENSITIVITY OF WINDOW LENGTH

We investigate the sensitivity of window length of some representative methods: SegRNN, DCRNN-dist, TSD, CrossFormer, and DSN. The results reported in Figure 7 show that in most cases the detection performance is positively correlated with the window length. In the extreme case where the window size is 1-s, the performance is on average -25.9% and -19.2% worse than the performance with 30-s windows respectively on FDUSZ and TUSZ datasets. Therefore, in the real-time detection task, it is impossible to use only the current EEG slice $\mathbf{X}_t \in \mathbb{R}^{S \times C}$ to precisely detect seizure events without historical information. In such cases, DSN is able to condense the historical information into memory banks and retrieve it with constant memory usage and time cost, while achieving state-of-the-art performance.

## A.8 SENSITIVITY OF DECISION THRESHOLD

Figure 8 provides the performance of all methods with respect to different decision thresholds, ranging from 0 to 1. It shows that the output of different models may have different distributions and a fixed threshold may bias the decision of seizure/non-seizure decisions. Therefore, in this paper, we use a threshold search to find the best threshold that has the highest $F_1$ score in the validation set.

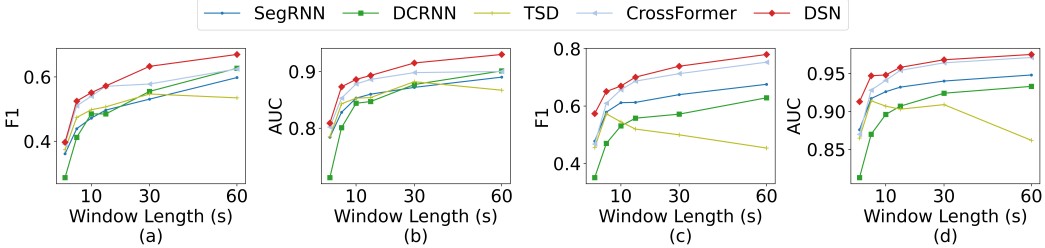

Figure 7: Sensitivity analysis of window length. (a)-(b) and (c)-(d) respectively show performance with different window length on FDUSZ and TUSZ datasets.

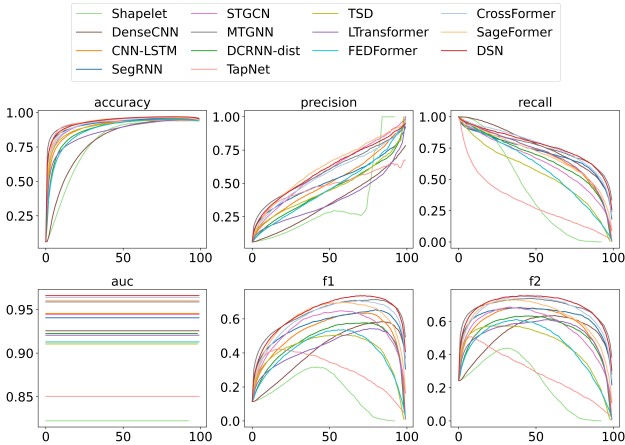

Figure 8: Sensitivity analysis of decision threshold of all methods.

## A.9 DEDUCTION OF THE RECURRENT ATTENTION MECHANISM

In this section, we aim to prove that the recurrent attention mechanism can be deduced from the self-attention operation. We begin with the typical self-attention operation with casual masking:

$$\mathbf{O}_i = \frac{\sum_j^i sim(\mathbf{Q}_i, \mathbf{K}_j)\mathbf{V}_j}{\sum_j^i sim(\mathbf{Q}_i\mathbf{K}_j)}, \tag{20}$$

where $sim(q, k) = exp(\frac{q^T k}{\sqrt{d}})$. Katharopoulos et al. (2020) relaxed the constraint of $sim(\cdot)$ and simplified the above equation to:

$$\mathbf{O}_i = \frac{\phi(\mathbf{Q}_i)\sum_j^i \phi(\mathbf{K}_j)^T \mathbf{V}_j}{\phi(\mathbf{Q}_i)\sum_j^i \phi(\mathbf{K}_j)^T}, \tag{21}$$

where $\phi(\cdot)$ is the kernel function.

We further introduce a trainable relative position encoding vector into the keys and use $exp(\cdot)$ as the kernel function:

$$\mathbf{O}_i = \frac{exp(\mathbf{Q}_i)\sum_j^i exp(\mathbf{K}_j + \mathbf{w}_{i-j})^T \mathbf{V}_j}{exp(\mathbf{Q}_i)\sum_j^i exp(\mathbf{K}_j + \mathbf{w}_{i-j})^T} \tag{22}$$

We define $\mathbf{S}_i = \sum_j^i exp(\mathbf{K}_j + \mathbf{w}_{i-j})^T \mathbf{V}_j$, which can be formulated in a recursive format:

$$\mathbf{S}_{i+1} = \sum_{j}^{i+1} exp(\mathbf{K}_j + \mathbf{w}_{i+1-j})^T \mathbf{V}_j \tag{23}$$

$$= \sum_{j}^{i} exp(\mathbf{K}_j + \mathbf{w}_{i+1-j})^T \mathbf{V}_j + exp(\mathbf{K}_{i+1} + \mathbf{w}_0)^T \mathbf{V}_{i+1} \tag{24}$$

$$= \sum_{j}^{i} \left(exp(\mathbf{w}_{i+1-j}) \odot exp(\mathbf{K}_j)\right)^T \mathbf{V}_j + \left(exp(\mathbf{w}_0) \odot exp(\mathbf{K}_{i+1})\right)^T \mathbf{V}_{i+1} \tag{25}$$

$$= \sum_{j}^{i} \left(\hat{\mathbf{w}}_{i+1-j} \odot exp(\mathbf{K}_j)^T\right) \mathbf{V}_j + \left(\hat{\mathbf{w}}_0 \odot exp(\mathbf{K}_{i+1})^T\right) \mathbf{V}_{i+1} \tag{26}$$

Assume $\hat{\mathbf{w}}_{i+1} = \mathbf{a} \odot \hat{\mathbf{w}}_i + \mathbf{b}, \forall i \geq 0$, then

$$\mathbf{S}_{i+1} = \mathbf{a} \odot \sum_{j}^{i} \left(\hat{\mathbf{w}}_{i-j} \odot exp(\mathbf{K}_j)^T\right) \mathbf{V}_j + \mathbf{b} \odot \sum_{j}^{i} exp(\mathbf{K}_j)^T \mathbf{V}_j + \left(\hat{\mathbf{w}}_0 \odot exp(\mathbf{K}_{i+1})^T\right) \mathbf{V}_{i+1} \tag{27}$$

$$= \mathbf{a} \odot \mathbf{S}_i + \mathbf{b} \odot \sum_{j}^{i} exp(\mathbf{K}_j)^T \mathbf{V}_j + \left(\hat{\mathbf{w}}_0 \odot exp(\mathbf{K}_{i+1})^T\right) \mathbf{V}_{i+1} \tag{28}$$

Let $\mathbf{P}_i = \sum_j^i exp(\mathbf{K}_j)^T \mathbf{V}_j$, we have

$$\begin{cases} \mathbf{S}_{i+1} = \mathbf{a} \odot \mathbf{S}_i + \mathbf{b} \odot \mathbf{P}_i + \left(\hat{\mathbf{w}}_0 \odot exp(\mathbf{K}_{i+1})^T\right) \mathbf{V}_{i+1} \\ \mathbf{P}_{i+1} = \mathbf{P}_i + exp(\mathbf{K}_{i+1})^T \mathbf{V}_{i+1} \end{cases} \tag{29}$$

Similarly, we define normalization factor $\mathbf{Z}_i = \sum_j^i exp(\mathbf{K}_j + \mathbf{w}_{i-j})^T$ and $\mathbf{Q}_i = \sum_j^i exp(\mathbf{K}_j)^T$ and represent them in recursive format:

$$\begin{cases} \mathbf{Z}_{i+1} = \mathbf{a} \odot \mathbf{Z}_i + \mathbf{b} \odot \mathbf{Q}_i + \left(\hat{\mathbf{w}}_0 \odot exp(\mathbf{K}_{i+1})^T\right) \\ \mathbf{Q}_{i+1} = \mathbf{Q}_i + exp(\mathbf{K}_{i+1})^T \end{cases} \tag{30}$$

Overall, all intermediate variables $\mathbf{S}_i$, $\mathbf{Z}_i$, $\mathbf{P}_i$ and $\mathbf{Q}_i$ can be formulated in a recursive fashion and be updated with query $\mathbf{Q}_{i+1}$, key $\mathbf{K}_{i+1}$ and value $\mathbf{V}_{i+1}$. We parameterize time decay factor $\mathbf{a}$, bias $\mathbf{b}$ and position encoding $\hat{\mathbf{w}}_0$ with neural networks. In Section 2.2.2, we let $\mathbf{b} = \mathbf{0}$ and ignore $\mathbf{P}_i$ and $\mathbf{Q}_i$ for simplicity.

The output of the recurrent attention network can be denoted as:

$$\mathbf{O}_i = \frac{exp(\mathbf{Q}_i)\mathbf{S}_i}{exp(\mathbf{Q}_i)\mathbf{Z}_i}. \tag{31}$$

## A.10 Random Samples

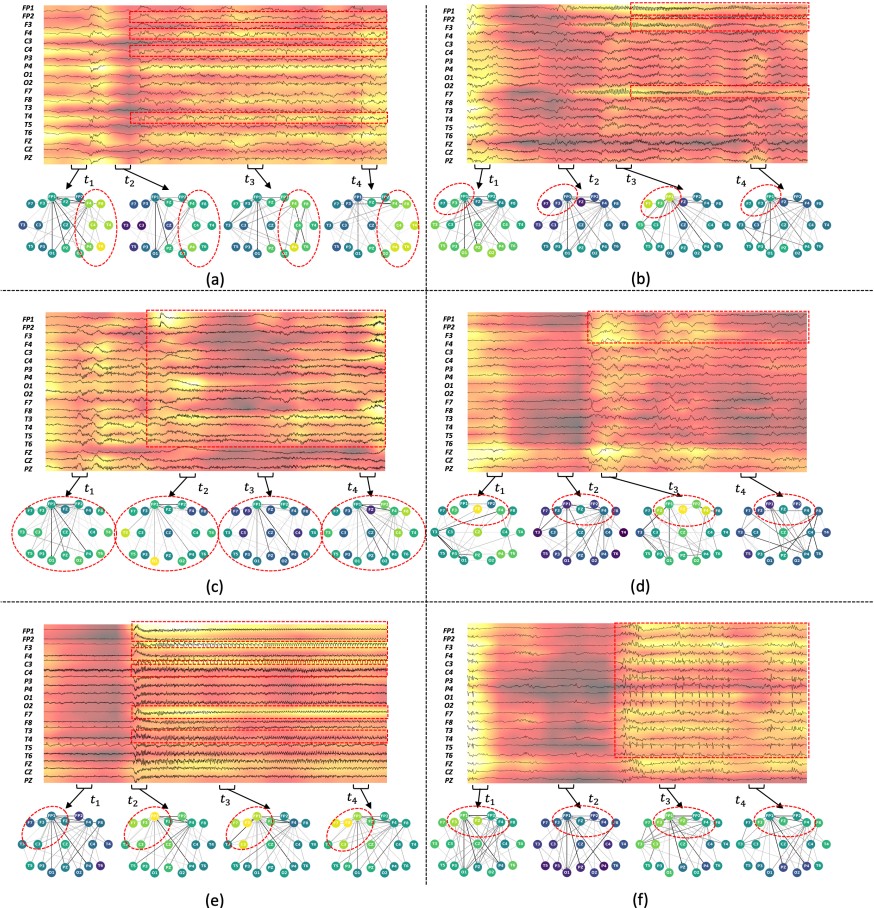

Figure 9: Visualization of some random samples with seizures. Red circles indicate the possible abnormal brain regions. Red boxes refer to annotated seizure labels.

## A.11 Limitations

Despite our best efforts to produce a comprehensive work, there are still many limitations in this paper and we decide to leave them in the future. (1) The visualization results of Figure 6 and Figure 9 have yet to be scrutinized by board-certified neurologists or relevant experts. (2) The mining longer temporal dependency on the real-time detection task remains to be investigated. (3) Several further studies towards the real-time detection task such as seizure duration prediction can be addressed in the future.

