# OpenReview forum: "Dynamic Electroencephalography Representation Learning for Improved Epileptic Seizure Detection"
_ICLR.cc/2024/Conference — Submitted to ICLR 2024_

### Official Review · Reviewer_Rq1a · 2023-10-21

**Soundness:** 3 good
**Presentation:** 3 good
**Contribution:** 2 fair
**Rating:** 5
**Confidence:** 4

**Summary:**

This paper proposes 1) a seizure onset detection task, 2) two metrics to quantify the timeliness of the detection methods, and 3) a Dynamic Seizure Network (DSN) to model the evolutionary brain states and dynamic brain connectivity from EEGs. Experimental results suggest that the proposed network outperforms baselines, and results in low time and space complexity. The authors further provide visualizations to display abnormal brain locations.

**Strengths:**

1. Technically sound paper.
2. Extensive experiments were performed and results were strong.

**Weaknesses:**

1. The proposed seizure onset detection task is not new. Here are a few examples of machine learning studies for seizure onset detection:
    * Lee, K., Jeong, H., Kim, S., Yang, D., Kang, H.-C., & Choi, E. (07--08 Apr 2022). Real-Time Seizure Detection using EEG: A Comprehensive Comparison of Recent Approaches under a Realistic Setting. In G. Flores, G. H. Chen, T. Pollard, J. C. Ho, & T. Naumann (Eds.), Proceedings of the Conference on Health, Inference, and Learning (Vol. 174, pp. 311–337). PMLR.
    * Tang, F.-G., Liu, Y., Li, Y., & Peng, Z.-W. (2020). A unified multi-level spectral–temporal feature learning framework for patient-specific seizure onset detection in EEG signals. Knowledge-Based Systems, 205, 106152.
    * M. Shama, D., Jing, J., & Venkataraman, A. (2023). DeepSOZ: A Robust Deep Model for Joint Temporal and Spatial Seizure Onset Localization from Multichannel EEG Data. Medical Image Computing and Computer Assisted Intervention – MICCAI 2023, 184–194.
2. Seizure involves long-range temporal dependency (e.g., 30 second). The proposed onset detection task uses very short window length (e.g., 1s) as inputs, which could be a limitation and prevent the model from learning long-range temporal dependency in EEGs.
3. The proposed metrics (diagnosis rate and wrong rate) only measure the correctness of predicting the start of the seizure, but do not measure the correctness of the predicted duration of the seizure.
4. Baseline selection seems a bit arbitrary. The models mentioned in Appendix Section A3 are seizure detection models and are highly related to this paper, but not all of them are chosen as baselines. e.g., BrainNet (Chen2022) and GraphS4mer (Tang2023) are not included as baselines. In particular, GraphS4mer also captures dynamic brain connectivity, so it would be good to compare DSN to it.
5. No details about hyperparameter selection is provided.

**Questions:**

1. As mentioned above, seizure onset detection is not a new task. Please conduct a thorough literature review and cite prior works.
2. What’s B in equation 3? How was B selected?
3. Please include a more comprehensive list of baselines, such as BrainNet and GraphS4mer. If a baseline cannot be included due to availability of open sourced code, please specify in the paper.
4. Please provide details about hyperparameter selection.
5. Were the examples in Figure 6 and Appendix Figure 9 reviewed by clinical experts? If not, please acknowledge it as a limitation.
6. As mentioned above, the seizure onset task does not capture long-range dependency in EEG during model training. Please acknowledge this limitation. A more optimal setup may be to train the models using longer window lengths, but predict seizure/non-seizure label within a short window (e.g., 1s) at inference time.
7. Please acknowledge the limitation that the proposed metrics do not take into account the predicted seizure duration.

---

> ### Author Response · Authors · 2023-11-20
> **Response to Reviewer Rq1a (part 1)**
>
> 1. *As mentioned above, seizure onset detection is not a new task. Please conduct a thorough literature review and cite prior works.*
>
> Thank you for the comment. After conducting a thorough review, we find that the previous tasks are fundamentally different from our task. Here is the reason.
>
> Firstly, we summarize the main characteristics of our onset detection task, which include:
>
> + **Streaming inputs/outputs**. Short EEG samples are input consecutively in a chronological order, and the model predicts corresponding seizure/non-seizure labels.
> + **Causal restriction**. The prediction at a given time step should only depend on the previous timesteps and not on future ones. This reflects the natural flow of temporal information.
> + **Long-range modeling**. Short samples are taken at distinct time points without any overlaps. Due to the relatively short duration of each sample, the main focus of this task is to model the long-range temporal correlations across consecutive samples.
>
> Secondly, after conducting a thorough literature review of previous works related to seizure onset detection, we have observed that these works fail to meet all the above criteria and have fundamentally different perspectives towards EEG signals.
>
> + The studied tasks in (Meier,2008) (Mansouri,2019) (Tang,2020, the second sample in your review) (Li,2021) (Wang,2021) (Maheshwari, 2022) (Shen,2023) (Saab,2023) only have literal similarities to our onset detection task and are fundamentally conventional seizure detection task. They use individual samples as inputs and model the *temporal dependency within each sample*, rather than across consecutive samples. Therefore, these studies do not fulfill the requirements of streaming characteristics.
> + The task described in (Shama, 2023, the third sample in your review) involves using non-overlapped samples with distinct timestamps as inputs and producing a label for each timestamp. However, this study only focuses on accurate predictions for each timestamp while *neglecting the causal restriction*. The proposed method in this study accesses information from the future timestamps with bidirectional LSTM modules, leading to information leakage. This limitation may restrict its applicability in real-time contexts. Therefore, this task has a fundamentally different perspective from our onset detection task, where the causality is preserved.
> + The study in (Lee, 2022, the first sample in your review) uses streaming EEG inputs/outputs that fulfill the causal restriction. However, the primary objective of this study is not to model the long-range dependency of EEG signals. Instead, it focuses on investigating methods to capture the *internal* spatial-temporal dependencies within each sample. They use overlapped samples with longer look-back windows to predict the seizure/non-seizure labels and use complex architectures like ResNet and DenseNet to model the local dependencies within each sample. In contrast, our task formulation simplifies the internal dependencies by utilizing shorter input samples and places a greater emphasis on modeling of *the long-range temporal dependency across consecutive samples*.
>
> In summary, while certain existing works may share some nominal similarities with our proposed onset detection task, they fail to meet all three essential characteristics and have fundamentally different perspectives towards EEG signals. Therefore, the novelty of our seizure onset detection task can be demonstrated.
>
> Moreover, we also propose two novel metrics, namely diagnosis rate and wrong rate, to evaluate the accuracy and latency of predictions simultaneously. Compared with the MARGIN metric proposed in (Lee,2022), which can only quantify the latency with a fixed time range, our metrics offer a more comprehensive evaluation of latency with different time ranges (denoted as $k$ in Equation 1 and 2). This allows for a more detailed assessment of prediction performance.
>
> 2. *What’s B in equation 3? How was B selected?*
>
> We apologize for the confusion. B refers to the length of blocks. In section 2.2.1, we split the EEG inputs into blocks with length B and apply the Fourier transformation to each block to extract the spectral features. Following (Tang,2022), we set B to 1 second, corresponding to 100 observed values within that time interval. We have updated the explanation of B in Equation 3.

---

> ### Author Response · Authors · 2023-11-20
> **Response to Reviewer Rq1a (part 2)**
>
> 3. *Please include a more comprehensive list of baselines, such as BrainNet and GraphS4mer. If a baseline cannot be included due to availability of open sourced code, please specify in the paper.*
>
> Thanks for your suggestion. We select the most representative methods and recent state-of-the-art methods for both multivariant time-series modeling or seizure detection from EEG signals as our baselines in our study. Although BrainNet (Chen, 2022) and GraphS4mer (Tang, 2023) are insightful state-of-the-art methods, we have decided to exclude them from the paper due to reproducibility concerns. We have updated the specific reasons for excluding them in the revised paper.
>
> Specifically, the open-source code for BrainNet is currently available. For GraphS4mer, we utilize the official implementation and hyperparameters obtained from the repository at https://github.com/tsy935/graphs4mer/ and install the acceleration kernels. However, despite our best efforts to carefully tune the hyperparameters, we are unable to reproduce the superior performance reported in the original paper. The obtained seizure detection performance of GraphS4mer on all datasets is shown in Table 1.  Among all methods, GraphS4mer ranks on average 10.3/7.0/1.7 on all three datasets under transductive setting and ranks 13.7/12.3 on FDUSZ/TUSZ datasets under inductive setting. The obtained results do not align with the superior performance described in the original paper. As a result, we have made the decision to exclude GraphS4mer from the baselines and will continue to investigate the reasons behind this discrepancy in the future.
>
> Furthermore, GraphS4mer exhibits longer training and inference times, as well as higher memory consumption. On our machine, the average inference time of GraphS4mer with 30-second window is 238.5s, which is 42 times slower than our DSN and 15 times slower than the average of other baselines. Additionally, during the inference stage, GraphS4mer consumes 52240MB GPU memory with a batch size of 256, which is 39 times larger than DSN with the same batch size and 25 times larger than the average of other baselines. This considerable inefficiency not only limits us to further investigate the capabilities of GraphS4mer but also raises concerns about its practical applicability in clinical contexts. Compared with GraphS4mer, our DSN has significantly better efficiency, thanks to the effective block-wise spectral embedding module, which efficiently extracts EEG features and significantly reduces the sequence length.
>
> We also compare our DSN with DeepSOZ (Shama,2023), one of the state-of-the-arts in Table 1. We use the official implementation of DeepSOZ available at https://github.com/deeksha-ms/DeepSOZ and conduct a grid-search to find the optimal hyperparameters for all datasets. Our DSN consistently outperforms DeepSOZ on all datasets. This improvement can be attributed to the ability to effectively localize brain abnormalities using the recurrent attention module and capture the dynamics of brain correlations.
>
> **Table 1. Seizure detection performance comparison of DSN, GraphS4mer and DeepSOZ. #rank refers to the rank of the performance among all baselines including GraphS4mer and DeepSOZ. #avg. rank refers to the average rank of all metrics on all datasets.**
>
> | Method     | Transductive |       |       |       |       |       |       |       |       | Inductive |       |       |       |       |       | #avg. rank |
> | ---------- | ------------ | ----- | ----- | ----- | ----- | ----- | ----- | ----- | ----- | ----- | ----- | ----- | ----- | ----- | ----- | ---- |
> |            | FDUSZ        |       |       | TUSZ |       |       | CHBMIT |       |       | FDUSZ |       |       | TUSZ |       |       |      |
> |            | F1           | F2    | AUC   | F1    | F2    | AUC   | F1    | F2    | AUC   | F1    | F2    | AUC   | F1    | F2    | AUC   |      |
> | DSN | 0.633 | 0.620 | 0.915 | 0.739 | 0.727 | 0.968 | 0.611 | 0.570 | 0.949 | 0.567 | 0.545 | 0.841 | 0.613 | 0.644 | 0.900 |      |
> | #rank | 1 | 1 | 1 | 1 | 1 | 1 | 2 | 1 | 1 | 1 | 1 | 1 | 1 | 1 | 1 | 1.1 |
> | GraphS4mer | 0.577 | 0.560 | 0.876 | 0.633 | 0.646 | 0.946 | 0.613 | 0.559 | 0.940 | 0.439 | 0.424 | 0.764 | 0.354 | 0.516 | 0.822 |      |
> | #rank | 7 | 12 | 12 | 8 |7 | 6 | 1 | 2 | 2 | 14 | 15 | 12 | 16 | 11 | 10 | 9.0 |
> | DeepSOZ | 0.557 | 0.560 | 0.884 | 0.606 | 0.609 | 0.936 | 0.413 | 0.405 | 0.890 | 0.405 | 0.416 | 0.744 | 0.421 | 0.525 | 0.832 |      |
> | #rank | 10 | 10 | 8 | 9 | 9 | 9 | 12 | 11 | 10 | 15 | 16 | 15 | 10 | 8 | 10 | 10.8 |
>
> 4. *Please provide details about hyperparameter selection.*
>
> Thank you for the comment. We describe the details about hyperparameter selection in the response to reviewer P4gj.

---

> ### Author Response · Authors · 2023-11-20
> **Response to Reviewer Rq1a (part 3)**
>
> 5. *Were the examples in Figure 6 and Appendix Figure 9 reviewed by clinical experts? If not, please acknowledge it as a limitation.*
>
> Thank you for the suggestion. The significance of Figure 6 and Appendix Figure 9 is to showcase that the hotspots are highly correspondence with the fluctuations of the EEG signals during the pre-ictal and ictal phases of seizures. Moreover, we have incorporated your suggestion and made the updates to Figure 6 and Appendix Figure 9. The red boxes have been added to indicate the annotated seizure labels, which showcases the strong alignment between the hotspots and the ground truth labels. These case studies demonstrate the DSN's ability to extract EEG features, localize active brain waves, and highlight the interpretability of our approach.
>
> However, it is important to note that the displayed heat maps have yet to undergo scrutiny by clinical experts. This is acknowledged as a limitation of our work, and we recognize the urgent need for clinical expert evaluation to further validate and refine the findings.
>
> 6. *As mentioned above, the seizure onset task does not capture long-range dependency in EEG during model training. Please acknowledge this limitation. A more optimal setup may be to train the models using longer window lengths, but predict seizure/non-seizure label within a short window (e.g., 1s) at inference time.*
>
> Thank you for the comment. The seizure onset task is able to capture the long-range dependency within EEG. Specifically, for streaming methods with memory banks such as DCRNN-dist and DSN, the historical information can be condensed and stored in the memory banks. During the training procedure, the losses of all timestamps are aggregated and back-propagated together. Therefore, the long-term dependency can be captured in the training procedure. During the inference procedure, the models retrieve historical information from the memory banks and make predictions within a short window (e.g., 1 second).
>
> The experiemental results can support this point of view. Figure 7 in Appendix A7 demonstrates that increasing the window length will benefit the capturing of long-term dependency within EEG signals and lead to better performance in most cases. Using short windows (1 second) will result in a catastrophic decrease in performance. However, in Figure 3, some streaming methods such as DSN, whose inputs are short 1-second clips, outperform other methods with longer inputs such as CrossFormer, whose inputs are 15-second clips. This result strongly proves that streaming methods are able to capture the long-range dependency within EEG by storing the historical information in the memory banks during the onset detection task and demonstrates the seizure onset task's ability to capture long-range dependency in EEG signals.
>
> 7. *Please acknowledge the limitation that the proposed metrics do not take into account the predicted seizure duration.*
>
> Thank you for the comment. Both seizure onset detection and duration prediction are important tasks in the diagnosis and management of epilepsy. Our main focus is to predict the start of seizure events, which has significant clinical implications. Prompt detection of seizure onset allows clinicians to intervene medically before the seizure becomes generalized or severe, thus improving patient outcomes (Meier, 2008) (Shoeb, 2010). While seizure duration prediction is certainly an important related task, we have chosen to leave it for future work and concentrate on seizure onset detection in this paper.
>
> We have summarized the limitations of this manuscript in the Appendix A11 in the revised paper.

---

> ### Author Response · Authors · 2023-11-20
> **Response to Reviewer Rq1a (part 4)**
>
> Reference:
> + Meier Ralph, et al. Detecting Epileptic Seizures in Long-term Human EEG: A New Approach to Automatic Online and Real-Time Detection and Classification of Polymorphic Seizure Patterns. In Journal of Clinical Neurophysiology, 2008.
> + Shoeb, Ali H. and John V. Guttag. Application of Machine Learning to Epileptic Seizure Detection.” In ICML, 2010.
> + Mansouri, Amirsalar et al. Online EEG Seizure Detection and Localization. In Algorithms, 2019.
> + Tang, Fang-Gui et al. A unified multi-level spectral-temporal feature learning framework for patient-specific seizure onset detection in EEG signals. In KBS, 2020.
> + Li, Chaosong et al. Seizure Onset Detection Using Empirical Mode Decomposition and Common Spatial Pattern. In IEEE Transactions on Neural Systems and Rehabilitation Engineering, 2021.
> + Wang Xiaoshuang, et al. One Dimensional Convolutional Neural Networks for Seizure Onset Detection Using Long-term Scalp and Intracranial EEG. In Neurocomputing, 2021.
> + J. Maheshwari, et al. Real-Time Automated Epileptic Seizure Detection by Analyzing Time-Varying High Spatial Frequency Oscillations. In IEEE Transactions on Instrumentation and Measurement, 2022.
> + Lee, Kwanhyung et al. Real-Time Seizure Detection using EEG: A Comprehensive Comparison of Recent Approaches under a Realistic Setting. In CHIL, 2022.
> + Chen, J. et al. BrainNet: Epileptic Wave Detection from SEEG with Hierarchical Graph Diffusion Learning. In KDD,2022.
> + Shen, Min et al. Real-time epilepsy seizure detection based on EEG using tunable-Q wavelet transform and convolutional neural network. In BSPC, 2023.
> + Saab, Khaled et al. Towards trustworthy seizure onset detection using workflow notes. In arxiv, 2023.
> + Shama, Deeksha M. et al. DeepSOZ: A Robust Deep Model for Joint Temporal and Spatial Seizure Onset Localization from Multichannel EEG Data. In MICCAI, 2023.
> + Tang, S., et al. Modeling Multivariate Biosignals with Graph Neural Networks and Structured State Space Models. In CHIL, 2023.

---

> > ### Comment · Reviewer_Rq1a · 2023-11-21
> >
> > Thank you for the responses.
> >
> > Re: 1. Instead of saying that you propose a new seizure onset task, please acknowledge prior seizure onset detection work, discuss their limitations, and why your framework addresses these limitations in Introduction.
> >
> > Re: 4, Learning rate is always an important hyperparameter and model performance can vary drastically between different learning rates. I’m surprised to see that learning rate is kept the same across models. Other hyperparameters listed in the table may not impact performance as much as learning rate.
> >
> > Re: 5, given that the visualizations are not interpreted by clinical experts, please be crystal clear about it in order not to mislead readers.

---

> > > ### Author Response · Authors · 2023-11-22
> > > **Response to Reviewer Rq1a (part 5)**
> > >
> > > *Re: 1. Instead of saying that you propose a new seizure onset task, please acknowledge prior seizure onset detection work, discuss their limitations, and why your framework addresses these limitations in Introduction.*
> > >
> > > Thanks for the suggestion. We have discriminated the seizure detection task, the onset detection, and the real-time detection task. We have also acknowledged prior related works, discussed the limitations of these works, and updated the description of the main contributions of our work in Introduction. To avoid confusion, we decide to modify the mentions of the task where inputs are streaming from `onset detection` to `real-time detection`.  We will update the related works in Appendix A2 later in the revised version of our manuscript.
> > >
> > >
> > >
> > > *Re: 4, Learning rate is always an important hyperparameter and model performance can vary drastically between different learning rates. I’m surprised to see that learning rate is kept the same across models. Other hyperparameters listed in the table may not impact performance as much as learning rate.*
> > >
> > > We apologize for the confusion. We acknowledge that using an identical learning rate for all methods is not very common. However, our choice of learning rate is not arbitrary. We have conducted a coarse-grained grid search on the learning rate for all methods on all datasets with search range [1e-4, 5e-4, 1e-3, 5e-3,1e-2]. Table 2 shows some cases of seizure detection performance on the TUSZ dataset under transductive with different learning rates. Surprisingly, we discover that 1e-3 is nearly the optimal learning rate for all methods. The reason might be that the Adam (Kingma,2014) optimizer and the cosine annealing scheduler (Loshchilov, 2017) help to stabilize the convergence. Moreover, some previous works also use the same learning rate across models after conducting a grid search. For example, (Tang,2022) uses an identical learning rate for all methods (i.e. CNN-LSTM, Dense-CNN, LSTM, DCRNN-dist, and DCRNN-corr) on seizure detection task after using grid-search to find the optimal learning rate. (Yu,2023) sets the learning rate to 1e-4 for all the methods on all the datasets. Therefore, we decide to use the same learning rate across models for convenience in this paper. We will conduct a more comprehensive grid search to tune the learning rate and provide a more detailed description of the experimental setting in the final version of our manuscript.
> > >
> > > **Table 2. Seizure detection performance of three case baselines on TUSZ dataset under transductive with different learning rates.**
> > >
> > > | Method      | lr       | F1        | F2        | AUC       |
> > > | ----------- | -------- | --------- | --------- | --------- |
> > > | DCRNN-dist  | 1e-2     | 0.537     | 0.550     | 0.912     |
> > > |             | 5e-3     | 0.566     | 0.574     | 0.923     |
> > > |             | **1e-3** | **0.572** | **0.596** | **0.924** |
> > > |             | 5e-4     | 0.571     | 0.592     | 0.922     |
> > > |             | 1e-4     | 0.529     | 0.558     | 0.908     |
> > > | CrossFormer | 1e-2     | 0.600     | 0.616     | 0.935     |
> > > |             | 5e-3     | 0.627     | 0.647     | 0.940     |
> > > |             | **1e-3** | 0.713     | 0.722     | **0.964** |
> > > |             | 5e-4     | **0.714** | 0.721     | 0.964     |
> > > |             | 1e-4     | 0.712     | **0.724** | 0.961     |
> > > | SageFormer  | 1e-2     | 0.599     | 0.598     | 0.935     |
> > > |             | 5e-3     | 0.670     | 0.673     | 0.955     |
> > > |             | **1e-3** | **0.711** | 0.704     | **0.964** |
> > > |             | 5e-4     | 0.708     | **0.704** | 0.963     |
> > > |             | 1e-4     | 0.609     | 0.624     | 0.939     |
> > >
> > >
> > >
> > > *Re: 5, given that the visualizations are not interpreted by clinical experts, please be crystal clear about it in order not to mislead readers.*
> > >
> > > Thanks for the comment. We have updated an announcement in Section 3.6 that the results require review and interpretation by clinical experts in real-world clinical settings. We also make an announcement that any insights derived from this study be subject to rigorous validation by board-certified neurologists or relevant experts in real-world clinical settings in Section Ethics Statement.
> > >
> > > Reference:
> > >
> > > + Loshchilov, I. and Hutter, F. SGDR: Stochastic Gradient Descent with Warm Restarts. In ICLR, 2017.
> > > + Kingma, D.P. and Ba, J. Adam: A Method for Stochastic Optimization. In ICLR, 2014.

---

> > > > ### Comment · Reviewer_Rq1a · 2023-11-22
> > > >
> > > > Thank you for your replies.

---

> > > > > ### Author Response · Authors · 2023-11-23
> > > > > **Thank you**
> > > > >
> > > > > Thank you for the review!

---

### Official Review · Reviewer_P4gj · 2023-11-01

**Soundness:** 2 fair
**Presentation:** 3 good
**Contribution:** 3 good
**Rating:** 3
**Confidence:** 4

**Summary:**

This paper proposes a new task called seizure onset detection that aims to identify the presence and timing of seizures from EEG signals. Existing seizure detection models is limited in real-world clinical use due to high latency and resource requirements, according to the authors. The paper also introduces two new metrics to quantify the detection latency.
The authors also propose a mode consisting of spectral embedding, evolutionary brain state modeling, and brain correlation learning.
Experiments are performed on three EEG datasets.

**Strengths:**

- Introduces a potentially useful seizure onset detection task and latency metrics.
- Achieves good results on the datasets and tasks, according to the paper's presentation.

**Weaknesses:**

My questions regarding the paper is primarily due to the experimental setup. We keep the basic settings of all baselines and our method to be identical (e.g. learning rate as 0.001, model dimension as 64) and use grid search to find the optimal model-specific hyper-parameters to get more reliable results, e.g. model layers and dropout rate. This is critical and does not posit a fair comparison. Other models were not designed for this architectural configuration or experimental setup. Its not clear what is model dimension. How were model parameters chosen for benchmarks? Was a grid-search performed for all of them and for both tasks (streaming and static) independently?

**Questions:**

Would like clarifications regarding weaknesses. Particularly the experimental setup of the evaluation

---

> ### Author Response · Authors · 2023-11-20
> **Response to Reviewer P4gj (part 1)**
>
> We apologize for the confusion. The model dimension refers to the dimension of hidden states, which is denoted by $D$ in this paper.
>
> We perform an exhaustive grid-search to find the optimal configurations of the hyperparameters independently for all baseline methods, for all datasets and for both tasks, in addition to following their official settings. The detailed strategies to choose parameters for all baselines are listed below and the searched ranges of all baselines are shown in the Table 2.
>
> + For baseline Shapelet, we scale the size of shapelets according to the length of EEG clips, leading to shapelets with length [10,20,40,80,160,320,640]. Grid-search is performed to find the best value of the number of shapelets.
> + For baseline DenseCNN, CNN-LSTM, DCRNN-dist and DCRNN-corr, we use the implementation from the open-source code of (Tang,2022) and utilize most of their hyperparameters. We perform grid-search on the number of layers and diffusion steps and dropout rate.
> + For baseline TSD, we set the hidden dimension to 64, in line with DCRNN-dist and DCRNN-corr and perform grid-search on other parameters, such as the number of attention heads and Transformer layers and dropout rate.
> + For other baselines such as MTGNN and CrossFormer, since they are not specially designed for seizure detection tasks or EEG datasets, we decide to use the same hidden dimension 64 as above. It is worth noting that using larger hidden dimension is not guaranteed for better performance. Taking CrossFormer as an example. Table 1 provides the performance with different hidden dimension on TUSZ dataset under transductive setting for seizure detection task. With an increased hidden dimension, the model appears to be overfitted and shows worse performance on the test set. Moreover, both the number of parameters (#Param.) and inference time (Inf. time) significantly increase. For other parameters, we use grid-search to find the optimal configurations for all datasets and for both tasks.
>
> It is worth noting that it is common to adopt an identical experimental setup for all baselines. For example, (Woo,2022) uses 320 as the representation dimensionality for all baselines, despite some of them are not specifically designed for the same task. (Yu,2023) sets the dimension of node memory to 172 and the dimension of time encoding to 100 for all methods across all the datasets. These studies also use identical loss function, optimizer, scheduler and learning rate configurations for all methods and all datasets, which are similar to our experimental setup. We have updated the description of experimental settings in Appendix A5.
>
> Reference:
>
> + Tang, S. et al. Self-Supervised Graph Neural Networks for Improved Electroencephalographic Seizure Analysis. In ICLR, 2022.
>
> + Woo, G. et al. CoST: Contrastive Learning of Disentangled Seasonal-Trend Representations for Time Series Forecasting. In ICLR, 2022.
>
> + Yu, L. et al. Towards Better Dynamic Graph Learning: New Architecture and Unified Library. In NeurIPS, 2023.

---

> ### Author Response · Authors · 2023-11-20
> **Response to Reviewer P4gj (part 2)**
>
> **Table 1. Comparison of seizure detection performance, parameter size and inference time of CrossFormer with different hidden dimension on TUSZ-Transductive. Results reported are the average over 5 runs with different random seeds.**
>
> | Method      | Hidden Dimension | F1    | F2    | AUC   | #Param. | Inf. Time |
> | ----------- | ---------------- | ----- | ----- | ----- | ------- | --------- |
> | CrossFormer | 32               | 0.696 | 0.701 | 0.960 | 158631  | 3.5s      |
> |             | 64               | 0.713 | 0.722 | 0.964 | 542631  | 5.1s      |
> |             | 128              | 0.707 | 0.696 | 0.962 | 2017191 | 6.0s      |
> |             | 256              | 0.697 | 0.679 | 0.958 | 7792551 | 13.2s     |
>
> **Table 2. Search ranges of hyperparameters of all methods. Grid search is performed independently for all datasets on both seizure detection task and onset task.**
>
> | Hyperparameter              | Searched  Ranges                                  | Related  Methods                                             |
> | --------------------------- | ------------------------------------------------- | ------------------------------------------------------------ |
> | Number  of layers           | $[1,2,3]$                                         | MTGNN,  SegRNN, DCRNN-dist, DCRNN-corr, TSD, LTransformer, FEDFormer, CrossFormer, SageFormer,  DSN |
> | Dropout  rate               | $[0.0,  0.1,0.2,0.3,0.4,0.5]  $                   | TapNet,  DenseCNN, STGCN, MTGNN, SegRNN, CNN-LSTM, DCRNN-dist, DCRNN-corr, TSD,  LTransformer, FEDFormer, CrossFormer, SageFormer, DSN |
> | Number  of attention heads  | $[1,2,4,8]$                                       | TSD,  LTransformer, FEDFormer, CrossFormer, SageFormer       |
> | Kernel  size                | $[3,5,9,13]$                                      | TapNet,  STGCN, MTGNN                                        |
> | Dilation                    | $[1,2,4,6]$                                       | TapNet,  STGCN, MTGNN                                        |
> | Number  of GCN layers       | $[1,2, 3]$                                        | MTGNN,  DCRNN-dist, DCRNN-corr, DSN                          |
> | Graph  normalization method | ['laplacian',  'random_walk', 'dual_random_walk'] | DCRNN-dist,  DCRNN-corr                                      |
> | Number  of shapelet         | $[3,5,7,9]$                                       | Shapelet                                                     |
> | propalpha                   | $[0,0.05,0.1,0.2]$                                | MTGNN                                                        |
> | decompose                   | $[True,False]$                                    | FEDFormer                                                    |
> | frequency  selection        | ['random','all']                                  | FEDFormer                                                    |
> | merge                       | $[1,2,4]$                                         | CrossFormer                                                  |
> | n_router                    | $[2,4,6,8,10]$                                    | CrossFormer                                                  |
> | n_cls_token                 | $[1,2,4]$                                         | SageFormer                                                   |
> | $\gamma$                    | $[1,0.2,0.1,0.05]$                                | DSN                                                          |
> | activation                  | ['tanh',  'relu', 'leakyRelu', None]              | DSN                                                          |

---

> ### Author Response · Authors · 2023-11-23
> **Response to Reviewer P4gj (part 3)**
>
> We would like to express our gratitude for your valuable suggestions in your previous review. We have carefully considered your review and have provided a detailed description of the experimental setting. We also provide the justification of learning rate selection in our response to reviewer Rq1a (part 5) at https://openreview.net/forum?id=5Gt68fnttu&noteId=KeEVVrf8wv.
>
> However, we have noticed that we have not yet received a response from you, and we are uncertain whether you have reviewed and approved the revised manuscript.
>
> Therefore, we kindly request a moment of your time to provide feedback on the revised version. Please feel free to contact us if you have further concerns.

---

### Official Review · Reviewer_UeqT · 2023-11-03

**Soundness:** 3 good
**Presentation:** 3 good
**Contribution:** 3 good
**Rating:** 6
**Confidence:** 4

**Summary:**

This paper presents a method, namely the onset detection, for seizure detection using EEG, and two metrics to quantify the timeliness of detection. The proposed method is based on a framework for EEG representation learning, which models the evolutionary brain states and dynamic brain connectivity efficiently. The experimental results conducted on two EEG databases have shown the effectiveness of the proposed method.

The paper is clear and well presented. The idea of the proposed contribution seems interesting.

**Strengths:**

- A method, called seizure onset detection task, for improving seizure detection task using EEG. The proposed method allows identifying the presence of seizures also the specific timestamps when seizures start.
- Two metrics were proposed (i.e., diagnosis rate and wrong rate) to quantify the timeliness of seizure detection under streaming context.
- A framework, called Dynamic Seizure Network (DSN), for efficient seizure detection from EEG signals that integrates the attention mechanism and the recurrent mechanism within a unified evolutionary state modeling module to localize brain abnormalities and model dynamic evolutionary patterns.

**Weaknesses:**

- Related work is not well described.
- The proposed method should be summarized as an algorithm allowing the readers to reproduce the proposed method.

**Questions:**

- It is suggested to discuss more the related work by considering some recent and relevant similar studies.
- It is also suggested to introduce an algorithm summarizing the proposed method.

---

> ### Author Response · Authors · 2023-11-20
> **Response to Reviewer UeqT**
>
> 1. *It is suggested to discuss more the related work by considering some recent and relevant similar studies.*
>
> Thanks for the suggestion. We have included more related works in the revised version in Appendix A1, including TIE-EEGNet (Peng,2022), MP-SeizNet (Albaqami,2023), GraphS4mer(Tang,2023) and DeepSOZ(Shama, 2023).
>
> Specifically, TIE-EEGNet enhanced the convolution layer of EEGNet (Lawhern, 2018) with sinusoidal encoding module. MP-SeizNet used a CNN module and a Bi-LSTM module with attention to learn two different EEG representations for seizure classification. GraphS4mer leveraged the Structured State Spaces architecture on raw EEG signals to capture long-range temporal dependencies and learned dynamically evolving graphs for biosignal modeling. DeepSOZ (Shama,2023) utilized attention and bi-LSTMs for precise seizure detection in each timestamp. Compared with the above methods, our DSN are more suitable for streaming EEG inputs with better performance and higher efficiency.
>
> We have updated the related works in Appendix A1.
>
> 2. *It is also suggested to introduce an algorithm summarizing the proposed method.*
>
> Thanks for the suggestion. We have included the algorithms of DSN on both seizure detection task and onset detection task in the Appendix A2 of the revised paper. Please check the revised paper for the algorithms since we are unable to upload images in this response.
>
> Reference:
>
> + Lawhern, V.J. et al. EEGNet: a compact convolutional neural network for EEG-based brain–computer interfaces. In Journal of Neural Engineering, 2018.
>
> + Chen, J. et al. BrainNet: Epileptic Wave Detection from SEEG with Hierarchical Graph Diffusion Learning. In KDD,2022.
>
> + Peng, R. et al. TIE-EEGNet: Temporal Information Enhanced EEGNet for Seizure Subtype Classification. In TNSRE, 2022.
> + Albaqami, H. et al. MP-SeizNet: A multi-path CNN Bi-LSTM Network for seizure-type classification using EEG. In BSPC, 2023.
> + Tang, S., et al. Modeling Multivariate Biosignals with Graph Neural Networks and Structured State Space Models. In CHIL, 2023.
> + Shama, Deeksha M. et al. DeepSOZ: A Robust Deep Model for Joint Temporal and Spatial Seizure Onset Localization from Multichannel EEG Data. In MICCAI, 2023

---

> > ### Comment · Reviewer_UeqT · 2023-11-20
> >
> > Thank you for addressing my comments.

---

> > > ### Author Response · Authors · 2023-11-22
> > > **Thank you**
> > >
> > > Thank you for the review!

---

### Author Response · Authors · 2023-11-20
**Response to AC and all reviewers**

We sincerely thank the reviewers for thoroughly examining this manuscript and providing valuable insights for the further improvements of this work. We have carefully considered these important suggestions and revised them accordingly. Please find our detailed responses to the reviewers' comments below. Additionally, we have made the following improvements to the paper:

+ Discrimination of the seizure detection/onset detection/real-time detection tasks, discussion of prior related works and description of the main contributions of our work are updated in Introduction.
+ To avoid confusion, we modify the mentions of the task where inputs are streaming from `onset detection` to `real-time detection`.
+ Description of the meaning of notation B in Equation 3 is added in Section 2.2.1.
+ Figure 6 and Appendix10 Figure 9 are updated with red boxes to indicate the annotated seizure ground truths as well as the description of these red boxes.
+ An annoucement hat the visualzation results require further reviewed by clinical experts in real-world clinical settings is updated in Section 3.6.
+ More related works are added in Appendix A1.
+ The pseudocode of DSN on seizure detection task and real-time detection task are shown in Appendix A2.
+ The explanation of the reproducibility issues of some related works is added in Appendix A3.
+ The description of experimental setting in Appendix A5 is updated for clarity.
+ The title of Figure 7 in Appendix A7 is updated for clarity.
+ The limitations of this manuscript are summarized in the Appendix A11.

---

### Meta-Review · Area_Chair_TxyS · 2023-12-10

**Metareview:**

The work targets the problem of epileptic seizure (onset) detection using deep learning proposing a framework called "Deep Seizure Network". Experiments are run on 3 different public datasets.

Features consist of the spectrogram (in dB) and the temporal network is a GRU with some modified self-attention mechanism. A block involving correlations is also added to capture "network" effects during seizures.

Evaluation is done either within subjects (same subjects in train and test) or across subjects. Algorithmic complexity and computation costs are discussed.

Overall, the contribution is primarily experimental with multiple comparisons of architectures. Many of these novel experiments have been added during rebuttal as appreciated by some reviewers, yet no reviewer has shown a strong enthusiasm on the novelty of the approach in terms of machine learning. Finally, questions remain on the fair evaluation of the comparison (choice of hyperparameters) and on their statistical significance.

Suggestions from AC:
- Adding error bars on metrics by doing multiple repeats between train and test splits would make the claims stronger.
- Use \max and define new commands in latex to avoid *italic* for words in latex equations.

**Justification For Why Not Higher Score:**

Limited machine learning contribution.
Limited scope of the contribution.

**Justification For Why Not Lower Score:**

Fair literature review.
Experimental evaluation comparing multiple deep learning methods.

---

### Decision · Program_Chairs · 2024-01-16

Reject